# Measurement report: Source apportionment of carbonaceous aerosol using dual-carbon isotopes ($^{13}C$ and $^{14}C$) and levoglucosan in three northern Chinese cities during 2018–2019

Huiyizhe Zhao [a, c, d], Zhenchuan Niu [a, b, c, d, e, *], Weijian Zhou [a, b, c, d, *], Sen Wang[f], Xue Feng[g], Shugang Wu [a, c], Xuefeng Lu [a, c], Hua Du [a, c]

[a] *State Key Laboratory of Loess and Quaternary Geology, CAS Center for Excellence in Quaternary Science and Global Change, Institute of Earth Environment, Chinese Academy of Sciences, Xi'an 710061, China*

[b] *Open Studio for Oceanic-Continental Climate and Environment Changes, Pilot National Laboratory for Marine Science and Technology (Qingdao), Qingdao 266061, China*

[c] *Shaanxi Provincial Key Laboratory of Accelerator Mass Spectrometry Technology and Application, Joint Xi'an AMS Center between IEECAS and Xi'an Jiaotong University, Xi'an 710061, China*

[d] *University of Chinese Academy of Sciences, Beijing 100049, China*

[e] *Shaanxi Guanzhong Plain Ecological Environment Change and Comprehensive Treatment National Observation and Research Station, Xi'an, China*

[f] *Shaanxi Key Laboratory of Earth Surface System and Environmental Carrying Capacity, College of Urban and Environmental Sciences, Northwest University, Xi'an, China*

[g] *Xi'an Institute for Innovative Earth Environment Research, Xi'an, China*

**Correspondence**: Zhenchuan Niu (niuzc@ieecas.cn) and Weijian Zhou (weijian@loess.llqg.ac.cn)

## Abstract

In this study, we investigated the characteristics and changes in the sources of carbonaceous aerosols in northern Chinese cities after the implementation of the Action Plan for Air Pollution Prevention and Control in 2013. We collected $PM_{2.5}$ samples from three representative inland cities, viz. Beijing (BJ), Xi'an (XA), and Linfen (LF) from January 2018 to April 2019. Elemental carbon (EC), organic carbon (OC), levoglucosan, stable carbon, and radiocarbon were measured in $PM_{2.5}$ to quantify the sources of carbonaceous aerosol employing Latin hypercube sampling. The best estimate of source apportionment showed that the emissions from liquid fossil fuels contributed 33.6 ± 12.9%, 26.6 ± 16.4%, and 24.6 ± 13.4% of the total carbon (TC) in BJ, XA, and LF, whereas coal combustion contributed 11.2 ± 9.1%, 19.2 ± 12.3%, and 39.2 ± 20.5%, respectively. Non-fossil sources accounted for 55 ± 11%, 54 ± 10%, and 36 ± 14% of the TC in BJ, XA, and LF, respectively. In XA, 48.34 ± 32.01% of non-fossil sources was attributed to biomass burning. The highest contributors to OC in LF and XA were fossil sources (65.4 ± 14.9% and 44.9 ± 9.5%, respectively), whereas that in BJ was non-fossil sources in BJ (56.1 ± 16.7%). The main contributors to EC were fossil sources, accounting for 92.9 ± 6.13%, 69.9 ± 20.9%, and 90.8 ± 9.9% of the total EC in BJ, XA, and LF, respectively. The decline (6–17%) in fossil source contributions in BJ and XA since the implementation of the Action Plan indicates the effectiveness of air quality management. We suggest that measures targeted to each city should be strengthened in the future.

**Keywords:** carbonaceous aerosols; radiocarbon; stable carbon; biomass; fossil fuel; source apportionment

## 1 Introduction

Atmospheric aerosols are extremely complex suspension systems. Carbonaceous aerosols are an important component of atmospheric aerosols, accounting for approximately 10–60% of the total mass of global fine particulate matter (Cao et al., 2003, 2007; Feng et al., 2009). Carbonaceous aerosols contain elemental carbon (EC), organic carbon (OC), and inorganic carbon (IC). IC is mainly derived from sand dust, it has a low concentration and simple composition, and it can be removed via acid treatment (Clarke et al., 1992). EC is produced by incomplete combustion and is directly discharged from pollution sources. It can cause global warming by changing the radiative forcing and ice albedo (Jacobson et al., 2001; Kiehl et al., 2007). OC is a complex mixture of primary and secondary pollutants produced by the combustion of domestic biomass and fossil fuels. It is an important contributor to tropospheric ozone, photochemical smog, and rainwater acidification, and it can significantly impact regional and global environments through biogeochemical cycling (Jacobson et al., 2000; Seinfeld et al., 1998). Therefore, identifying and quantifying the source contributions of carbonaceous aerosols can provide a scientific basis for the management of regional air quality.

The natural radiocarbon isotope ($^{14}C$) can be used to study the source of atmospheric particulate matter and to quantitatively and accurately distinguish the contributions of fossil and non-fossil sources (Clayton et al., 1955; Currie, 2000; Szidat, 2009). In recent decades, this method has been widely used to trace non-fossil carbonaceous aerosols in various regions (Ceburnis et al., 2011; Huang et al., 2010; Lewis et al., 2004; Szidat et al., 2009; Vonwiller et al., 2017; Yang et al., 2005; Yttri et al., 2011; Zhang et al., 2012, 2017). Stable carbon isotope ($^{13}C$) is useful geochemical marker that can provide valuable information about both the sources and atmospheric

processing of carbonaceous aerosols (López-Veneroni, 2009; Widory, 2006), and they have been applied in various types of environmental research to identify emission sources (Cachier et al., 1985, 1986; Cao et al., 2011; Chesselet et al., 1981; Fang et al., 2017; Kawashima & Haneishi, 2012; Kirillova et al., 2013; Liu et al., 2014; Wang et al., 2012). The analysis of $^{13}C/^{12}C$ can refine $^{14}C$ source apportionment because both coal and liquid fossil fuels are depleted of $^{14}C$ while their $^{13}C$ source signatures are different (Andersson et al., 2015; Li et al., 2016; Winiger et al., 2017). Levoglucosan (Lev), a thermal degradation product of cellulose combustion, is a common molecular tracer that can be used to evaluate the contribution of biomass burning (Hoffmann et al., 2010; Locker et al., 1988; Simoneit et al., 1999). The combination of the carbon isotope analysis and Lev can further divide the contributions of different carbonaceous sources. Some studies have confirmed the feasibility of this combination (Claeys et al., 2010; Gelencsér et al., 2007; Genberg et al., 2011; Huang et al., 2014; Kumagai et al., 2010; Liu et al., 2013; Niu et al., 2013; Zhang et al., 2015).

Cities in northern China have been affected by severe haze for several decades. After the Action Plan for Air Pollution Prevention and Control (hereafter simplified as "Action Plan") was promulgated in 2013, all parts of China responded to the issue and held numerous air quality management practices (CSC, 2013). In 2020, the average $PM_{2.5}$ concentration in Chinese cities across the country decreased by 54.2% compared to that in 2013 (MEE, 2014, 2021). In 2020, the proportion of clean energy consumption, such as that of natural gas and electricity, increased by 7.9% compared to that in 2013, and the proportion of coal combustion decreased by 9.7% (NBS, 2021). Before the Action Plan, fossil fuel sources were identified as the main contributor to carbonaceous aerosols in Chinese cities (56–81%) (Ni et al., 2018, Niu

et al., 2013, Shao et al., 1996; Sun et al., 2012; Yang et al., 2005). In this study, we aimed to determine the main contribution of the current carbonaceous aerosols in northern Chinese cities. Also, we aimed to identify whether changes in energy type and emission control caused a change in the source of carbonaceous aerosols.

To address those issues, we conducted a source apportionment of carbonaceous aerosols based on yearly measurements of OC, EC, Lev, $^{13}C$, and $^{14}C$ in $PM_{2.5}$, combined with Latin hypercube sampling (LHS), in three representative northern Chinese cities during 2018–2019. This study provides a comprehensive understanding of current sources of carbonaceous aerosol after the implementation of the Action Plan in Chinese cities.

## 2 Methods

### 2.1 Research sites

We selected one urban sampling site in Beijing (BJ), one in Xi'an (XA), and one in Linfen (LF) (Fig. 1). BJ is the capital of China, one of the largest megacities in the world, and the central city of the Beijing–Tianjin–Hebei economic region. It has a population of more than 20 million and has experienced serious air pollution problems in the past few decades. XA, the capital of Shaanxi Province, is the ninth-largest central city and an important city of the Northwest Economic Belt in China. It is located in a basin surrounded by mountains on three sides, where atmospheric pollutants are discharged mainly from the basin and are less affected by other urban areas (Cao et al., 2009; Shen et al., 2011). LF is located in western Shanxi Province and is one of the representative cities in the northern air-polluted region. Shanxi Province is the center of Chinese energy production and chemical metallurgy industries; its coal production and consumption were approximately 736.81 million

tons and 349.07 million tons, accounting for 27.05% and 12.42% of the Chinese total in 2019, respectively (NBS, 2020; SPBS, 2020). The air quality in LF was ranked in the worst ten in China from 2018 to 2020 (MEE, 2019, 2020, 2021). According to China Central Television (CCTV) reports, the atmospheric $SO_2$ concentration in LF exceeded 1000 μg $m^{-3}$ several times during January 2017 (CCTV, 2017). XA and LF heavily suffer from air pollution in the Fenwei Plain. In July 2018, the State Council issued the Three-Year Action Plan to Win the Blue Sky Defense War. this included the Fenwei Plain as one of the key areas in which to prevent and control pollution (CSC, 2018).

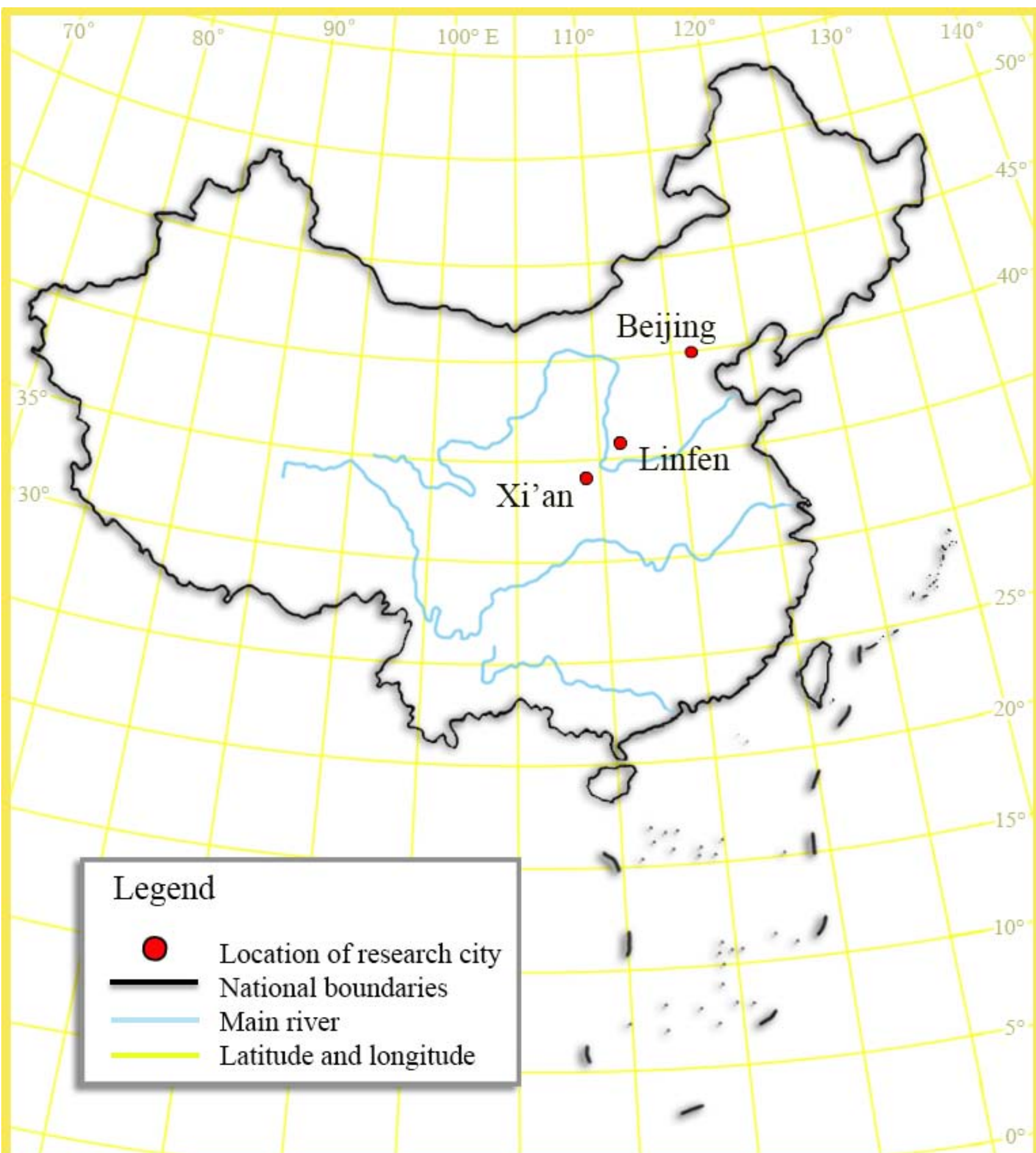

Fig. 1 Locations of Beijing (BJ), Xi'an (XA), and Linfen (LF).

The first site was located in the northwest of BJ, on the rooftop of the Research Center for Eco-Environmental Sciences, Chinese Academy of Sciences (40°0′33″ N, 116°20′38″ E). The site was approximately 200 m from the road. The second site was located southwest of XA, on the rooftop of the School of Urban and Environmental Sciences in Northwest University (34°15′36″ N, 108°88′53″ E). Living quarters and teaching areas were located around these two sampling sites. The third site was located in Houma, a county-level city of LF, on the rooftop of a residential building (35°63′56″ N, 111°39′53″ E). There was no industrial pollution near each site and they were representative urban sites.

**2.2 Sample collection**

At BJ and XA, $PM_{2.5}$, samples were collected on the 7$^{th}$, 14$^{th}$, 21$^{st}$, and 28$^{th}$ of each month from April 28, 2018, to April 21, 2019. In LF, seven consecutive days in each season were selected for sample collection, and the sampling periods were concentrated in January, April, July, and October 2018.

Samples in each city were collected continuously on pre-baked quartz fiber filters (203 mm × 254 mm, Whatman UK) using a high-volume (1.05 $m^3$ $min^{-1}$) sampler (TH-1000CII). To remove the existing carbon in the materials, the filter and foil used for wrapping should be baked in a muffle furnace at 375 °C for 5 h before use. After sampling, the filters were folded, wrapped in pre-baked aluminum foil, and stored at −18 °C until analysis. All filters were weighed after equilibrating at 25 ± 1 °C and 52 ± 5% humidity for more than 24 h. 124 $PM_{2.5}$ samples and 4 field blanks were obtained in total.

**2.3 OC and EC analyses**

Filter pieces of 0.526 $cm^2$ were used to measure the OC and EC using a DRI

Model 2001 (Thermal/Optical Carbon Analyzer) at the Institute of Earth Environment, Chinese Academy of Sciences. The Interagency Monitoring of Protected Visual Environments (IMPROVE) thermal/optical reflectance protocol must be followed because OC and EC have different oxidation priorities under different temperatures (Cao et al., 2007; Chow & Watson, 2002). OC and EC were defined as OC1 + OC2 + OC3 + OC4 + OP and EC1 + EC2 + EC3 − OP, respectively, in accordance with the IMPROVE protocol (Chow et al., 2004). Sample analysis results were corrected by the average blank and standard sucrose concentrations of OC and EC, respectively.

### 2.4 Lev analysis

The molecular tracer (Lev) was determined by high-performance anion exchange chromatography with pulsed amperometric detection (HPAEC-PAD) method at the South China Institute of Environmental Science, Ministry of Ecology and Environment. A quartz filter sample (2 $cm^2$) was extracted with 3 ml of deionized water in a prebaked glass bottle under ultrasonic agitation and was subsequently analyzed using a Dionex ICS-3000 system after filtration. The separation requires an equilibrium period, isocratic elution, and gradient elution. (For a specific description, refer to Zhang et al., 2013.) The instrument sample loop was 100 μL and the detection limit of Lev was $1\times10^{-8}$ μg $ml^{-1}$.

### 2.5 Stable carbon analysis

The $^{13}C$ compositions were determined using a gas isotopic analyzer (Picarro G2131-i) in conjunction with an elemental analyzer (Elemental Combustion System 4010) at the Institute of Earth Environment, Chinese Academy of Sciences. Specifically, 0.2–0.4 mgC of sample was placed in a precombusted tin capsule (6×10 mm) and the air was removed by squeezing. The samples were tested at 980 °C and 650 °C with 70–80 ml $min^{-1}$ helium as the carrier gas and 20–30 ml $min^{-1}$ oxygen as

the reaction gas. The resulting gas mixture was then collected in Gas Isotopic Analyzer. Urea standard (CAS Number: 57-13-6) was used as standard sample. $^{13}C$ data are expressed in delta notation with respect to Vienna Pee Dee Belemnite (VPDB) (Coplen, 1996):

$$\delta^{13}C = [\frac{^{13}C/^{12}C_{Sample}}{^{13}C/^{12}C_{VPDB}} - 1] \times 1000‰ \quad (1)$$

**2.6 Radiocarbon analysis**

The $^{14}C$ samples were prepared and tested in the laboratory of Xi'an accelerator mass spectrometer (AMS) Center. Carbonate must be removed from the filters using hydrochloric acid (1 M) before combustion. The processed sample was packed in a sealed quartz tube with a silver wire and excessive CuO. The solid sample was then combusted at 850 °C for 2.5 h to convert it into gas after the vacuum degree was less than $5\times10^{-5}$ mbar. The gas sample was passed through a liquid nitrogen cold trap (−196 °C) to freeze $CO_2$ and water vapor, and then passed through an ethanol–liquid nitrogen cold trap (−90 °C) to remove water vapor and purify $CO_2$ (Turnbull et al., 2007; Zhou et al., 2014). The collected $CO_2$ was reduced to graphite via a reduction reaction with zinc particles and iron powder as the reductant and catalyst, respectively (Jull, 2007; Slota et al., 1987). The graphite was pressed into an aluminum container and measured using a 3 MV AMS, with a precision of 3‰ (Zhou et al., 2006, 2007). Forty-nine targets were arranged in sequence in the sample fixed wheel, including fourty samples, six OX-II standard samples, two anthracite standard samples and one sugar carbon standard sample each time. AMS online $\delta^{13}C$ of was used for isotope fractionation correction.

The $^{14}C$ results were expressed as a fraction of modern carbon ($f_M$) (Currie, 2000; Mook & Van Der Plicht, 1999). It defines as the $^{14}C/^{12}C$ ratio of the sample related to the isotopic ratio of the reference year 1950 (Stuiver & Polach, 1977):

$$f_M = (^{14}C/^{12}C_{Sample})/(^{14}C/^{12}C_{1950}). \quad (2)$$

Non-fossil fractions ($f_{nf}$) and fossil fractions ($f_f$) were determined from the $f_M$ values.

$$f_{nf} = f_M \times 100\% \quad (3)$$

$$f_f = (1 - f_M) \times 100\% \quad (4)$$

### 2.7 Source apportionment of total carbon using $^{14}C$ and $^{13}C$

To study the contribution of each fossil source to the total carbon (TC), we used the principle of isotopic chemical mass balance to further separate fossil sources into liquid fossil fuels and coal. Since the amount of carbonaceous aerosol produced by natural gas is very low compared to coal and liquid fossil combustion, its contribution was not considered here (Chen et al., 2005; England et al.., 2002; Guo et al., 2014; Yan et al., 2010). In this part, $^{13}C$ and $^{14}C$ were combined to calculate the contributions of non-fossil, coal, and liquid fossil sources.

$$f_{nf} \times \delta^{13}C_{nf} + f_{coal} \times \delta^{13}C_{coal} + f_{liq.fossil} \times \delta^{13}C_{liq.fossil} = \delta^{13}C_{sample} \quad (5)$$

$$f_{coal} + f_{liq.fossil} = f_f \quad (6)$$

$\delta^{13}C_{sample}$ is the $\delta^{13}C$ of the samples at each site; $f_{nf}$, $f_{coal}$, and $f_{liq.fossil}$ represent the proportions of each source; and $\delta^{13}C_{nf}$, $\delta^{13}C_{coal}$, and $\delta^{13}C_{liq.fossil}$ represent $\delta^{13}C$ from the corresponding sources. The selection of the reference value is described in detail in Section 2.9.

### 2.8 Source apportionment of OC and EC using $^{14}C$ and Lev

The method combines $^{14}C$ with the concentration of carbon components and a molecular tracer (Lev) to quantify the sources of OC and EC. Carbon was assumed to originate from fossil fuel combustion, biomass burning, and other non-fossil emissions (Gelencsér et al., 2007). The following is a simple calculation method.

EC consists of biomass burning ($EC_{bb}$) and fossil fuel combustion ($EC_{ff}$).

$$EC = EC_{ff} + EC_{bb} \quad (7)$$

$EC_{bb}$ was calculated based on the Lev concentration and the estimated $EC_{bb}$/Lev ratio:

$$EC_{bb} = Lev \times (EC_{bb}/Lev) = Lev \times [(EC/OC)_{bb}/(Lev/OC_{bb})] \quad (8)$$

Then, $EC_{ff}$ was calculated by subtraction (Eq. 7).

OC consists of OC from biomass burning ($OC_{bb}$), fossil fuel combustion ($OC_{ff}$), and other sources ($OC_{other}$), including primary and secondary biogenic OC and SOC (secondary organic carbon) from non-fossil emissions.

$$OC = OC_{bb} + OC_{ff} + OC_{other} \quad (9)$$

$OC_{bb}$ was calculated based on the Lev concentration and the estimated Lev/$OC_{bb}$ ratio:

$$OC_{bb} = Lev/(Lev/OC_{bb}) \quad (10)$$

$OC_{other}$ was calculated by balancing the $^{14}C$ content that was not attributed to $OC_{bb}$.

$$OC_{other} = (OC \times f_{nf}(OC) - OC_{bb} \times f_M(bb))/ f_M(nf). \quad (11)$$

Furthermore, $f_{nf}$(OC) was calculated based on the $^{14}C$ concentration measured in the sample (detailed description of the formulas can be found in Genberg et al., 2011); $f_M$(bb) and $f_M$(nf) are the $^{14}C$ concentrations in biomass burning and other non-fossil emissions, respectively.

Finally, $OC_{ff}$ was calculated by subtraction (Eq. 9).

**2.9 Uncertainties of source apportionment**

Some uncertainties exist in some parameters in Eqs. 5–11 and need to be evaluated. Table 1 lists the range of reference values used in this study. The ratios Lev/$OC_{bb}$ and $EC_{bb}$/$OC_{bb}$ depend on the type of biofuel and the burning conditions (Oros et al., 2001a, b). In foreign studies, the most common distributions of Lev/$OC_{bb}$

and $EC_{bb}/OC_{bb}$ are 0.08–0.2 and 0.07–0.45, respectively (Gelencsér et al., 2007; Puxbaumet et al., 2007; Szidat et al., 2006). The distribution ranges of $Lev/OC_{bb}$ and $EC_{bb}/OC_{bb}$ burned by trees, shrubs, and rice are approximately 0.01–0.04 and 0.05–0.31, respectively (Engling et al., 2006, 2009; Wang et al., 2009). Zhang et al. (2007) found that the values of $Lev/OC_{bb}$ and $EC_{bb}/OC_{bb}$ in the cereal straw of BJ were 0.08 and 0.13, respectively.

**Table 1. Values with limits of input parameters for source apportionment using Latin hypercube sampling (LHS).**

| Parameters | Low | Probable value | High |
|---|---|---|---|
| $Lev/OC_{bb}$ | 0.01 | 0.08 | 0.20 |
| $EC_{bb}/OC_{bb}$ | 0.13 | 0.16 | 0.31 |
| $\delta^{13}C_{liq.fossil}$ (‰) | −28.00 | −27.00 | −25.00 |
| $\delta^{13}C_{Coal}$ (‰) | −25.00 | −22.95 | −21.00 |
| $\delta^{13}C_{nf}$ [a] (‰) | −26.00 | −25.25 | −24.00 |
| $\delta^{13}C_{nf}$ [b] (‰) | −27.00 | −26.50 | −25.00 |

Agnihotri et al., 2011; Engling et al., 2006, 2009; Gelencsér et al., 2007; Huang et al., 2006; Lopez-Veneroni, 2009; Martinelli et al., 2002; Moura et al., 2008; Oros et al., 2001a, b; Puxbaumet et al., 2007; Smith & Epstein, 1971; Szidat et al., 2006; Turekian et al., 1998; Wang et al., 2009; Widory, 2006; Zhang et al., 2007.

[a] Values used in BJ/LF

[b] Values used in XA

The $\delta^{13}C$ of aerosols derived from liquid fossil fuels (gasoline and diesel oil) was approximately −28 ‰ to − 25 ‰ (Agnihotri et al., 2011; Huang et al., 2006; Lopez-Veneroni, 2009; Widory, 2006). The $\delta^{13}C$ derived from coal combustion was relatively high, ranging from −25 ‰ to −21 ‰ (Agnihotri et al., 2011; Widory, 2006).

The results of Agnihotri et al. (2011) showed that the $\delta^{13}C$ characteristic of biomass burning emissions ranged from −25.9 ‰ to −29.4 ‰. Smith & Epstein (1971) found that plants with C3 (e.g., wheat, soybeans, and most woody plants) and C4 (e.g., corn, grass, and sugar cane) metabolism had significantly different $\delta^{13}C$, with an average of −27 ‰ and −13 ‰, respectively. In other studies, these two types of plant-derived aerosols had different characteristics; the $^{13}C$ from C3 and C4 plants ranged from approximately −23.9 ‰ to −32 ‰ (Moura et al., 2008; Turekian et al., 1998) and from −11.5‰ to −13.5 ‰ (Martinelli et al., 2002), respectively.

Because of the differences in the structure of biomass fuels in different cities, we selected the $\delta^{13}C$ value based on the current status of biomass fuel used in research regions. In China, biomass fuels mainly include crop residues, branches, and leaves, and the amount of perennial wood is quite small (Zhang et al., 2015). BJ has a small area of arable land, with low agricultural output and corn production (BJMBS, 2020). The neighboring province, Hebei, is a large agricultural province that produces a large amount of wheat and corn annually; the latter has a larger sown area (PGHP, 2020). Shanxi Province also mainly produces wheat and corn; however, the sown area of corn is more than three times that of wheat (SPBS, 2020). Agricultural production in XA and the surrounding Guanzhong area is relatively large. The agricultural structure is dominated by wheat and corn, and their sown areas are not very different (SAPBS, 2020). This shows that the $\delta^{13}C$ of agricultural straw burning in LF is likely to be higher and that in XA may be lower. Some studies considered that $\delta^{13}C$ used for quantitative mass–balance source apportionment calculations from biomass burning should mainly be defined as C3 plants (Anderson et al., 2015; Fang et al., 2017; Ni et al., 2020). Based on this information, the $\delta^{13}C$ value of biomass burning in BJ and LF was found to be approximately −26 ‰ to −24 ‰, and that in XA is likely to be from

approximately −27 ‰ to −25 ‰.

Nuclear bomb tests in the late 1950s and the early 1960s released a large amount of $^{14}C$, and the ratio of $^{14}C/^{12}C$ in atmospheric $CO_2$ roughly doubled in the mid-1960s (Hua & Barbetti, 2004; Levin et al., 2003, 2010; Lewis et al., 2004; Niu et al., 2021). However, $f_M$ in the atmosphere has been decreasing because of the dilution effect produced by the absorption of marine and terrestrial biospheres and the release of fossil fuels. In recent years, studies on background $^{14}CO_2$ in China and other countries have shown that the $f_M$ value in the atmosphere has decreased and approached 1 (Hammer et al., 2017; Niu et al., 2016). This means that the impact of the nuclear explosions has almost disappeared, and the current changes in atmospheric $^{14}C$ are mainly influenced by the regional natural carbon cycle and fossil fuel $CO_2$ emissions. As perennial biomass fuel is less frequently used in China, the impact of nuclear explosions on $^{14}C$ data can be ignored, and the $f_M(nf)$ and $f_M(bb)$ of the local station should be close to the atmospheric value.

To evaluate the uncertainties of the quantification of source contributions, which resulted from the uncertainties of the parameters used, we used Python software to generate 3000 random variable simulations based on the LHS method (Gelencsér et al., 2007). After excluding part of the out-of-range data, the median value of the remaining simulations of each sample was considered as the best estimate. The results of the uncertainties analysis is discussed further in Section 3.6.

**2.10 Air mass backward trajectory analysis**

For Backward trajectory analysis, air-mass back trajectories from the previous 48 h were determined by using the HYbrid Single-Particle Lagrangian Integrated Trajectory (HYSPLIT) model (Draxler and Hess, 1998) at three different endpoint heights (e.g., 100 m, 500 m, and 1000 m) and a time interval of 6 h for sampling day

(https://www.arl.noaa.gov/).

## 3 Results and discussion

### 3.1 Characteristics and variation of carbonaceous components

During the sampling period, the average mass concentration of $PM_{2.5}$ in BJ, XA, and LF was 72.07 ± 44.87, 98.61 ± 64.53, and 175.00 ± 134.41 μg $m^{-3}$, respectively. All concentrations were higher in winter and lower in summer; LF showed the highest value of 368.71 ± 74.96 μg $m^{-3}$ in winter.

Fig. 2 shows the changes in OC and EC and their ratios at the sampling sites. The carbon components in the BJ, XA, and LF samples accounted for approximately 17.5 ± 6.0%, 21.5 ± 21.0%, and 17.8 ± 7.2% of $PM_{2.5}$, respectively. Both OC and EC were changing simultaneously and were characterized by low carbonaceous concentrations in summer (OC: 8.85 ± 3.71 μg $m^{-3}$; EC: 1.56 ± 0.92 μg $m^{-3}$) and high concentrations in winter (OC: 69.22 ± 58.94 μg $m^{-3}$; EC: 11.81 ± 7.88 μg $m^{-3}$). The average OC/EC ratios in BJ, XA, and LF were 3.95 ± 1.41, 8.98 ± 6.09, and 6.58 ± 2.04, respectively. Recent studies have shown that the average ratio of OC/EC in BJ, XA, and Shanxi Province was approximately 1.22–6.5 (Han et al., 2016; Ji et al., 2018; Wang et al., 2015; Zhao et al., 2013). Generally, secondary OC (SOC) is considered to occur when OC/EC > 2 (Castro et al., 1999; Novakov et al., 2005; Turpin & Huntzicker, 1995). The high ratio indicates that all carbonaceous aerosols contained a large number of SOCs, especially in XA.

The average mass concentrations of TC, OC, and EC at the sampling site in BJ were 12.50 ± 11.79, 9.73 ± 9.99, and 2.77 ± 2.12 μgC $m^{-3}$. The concentration of carbon components was relatively stable in spring and summer but fluctuated greatly in autumn and winter. The concentration of carbon components in most cases was
close to that of other periods, but there was a rapid increase in autumn and winter. The highest TC value was observed in the middle of January 2019 (81.51 μgC $m^{-3}$).

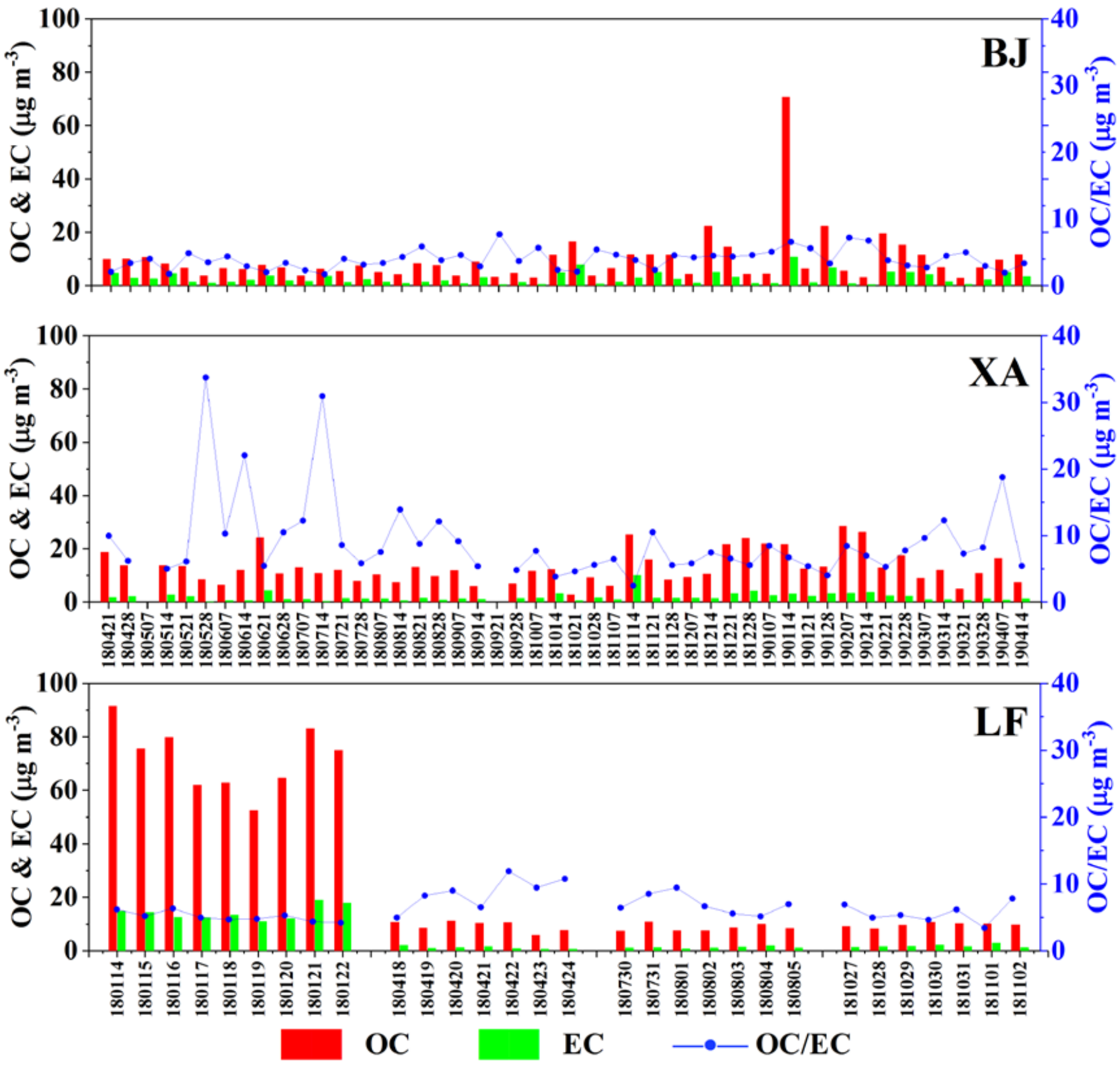

Fig. 2 Variations of carbon components and their ratios in $PM_{2.5}$ at the sampling sites in Beijing (BJ), Xi'an (XA), and Linfen (LF) (date, "yymmdd").

The average concentrations of TC, OC, and EC in XA were 14.64 ± 7.52, 12.76 ± 6.26, and 1.89 ± 1.61 μgC $m^{-3}$, respectively. In contrast to that in BJ, the concentration of the carbon components in XA fluctuated greatly throughout the year. Specifically, the concentration was lower from July to October and significantly higher from December to February. However, there were high concentrations of TC on some days in spring and summer, such as June 21, 2018, with the concentration

reaching 28.75 μgC $m^{-3}$.

The average concentrations of TC, OC, and EC in LF were 35.66 ± 36.53, 30.03 ± 30.40, and 5.64 ± 6.24 μgC $m^{-3}$, respectively. In contrast to those in BJ and XA, the concentration of the carbon components in LF was persistently high in winter and stable and low in other seasons.

### 3.2 Variations of $^{14}C$

The $^{14}C$ results showed that the average $f_{nf}$ values in BJ, XA, and LF were 55 ± 11%, 54 ± 10%, and 36 ± 14%, respectively. Non-fossil sources were the main contributors in the BJ and XA samples (Fig. 3). Furthermore, the $f_{nf}$ in the BJ samples showed a higher average value in spring (59 ± 6%), whereas that in the XA samples had higher average values in autumn ($f_{nf}$, 59 ± 7%) and winter ($f_{nf}$, 64 ± 6%). In the LF samples, fossil sources were the main contributors, contributing 80 ± 1% in winter.

By analyzing the $f_{nf}$ characteristics of samples with different pollution levels based on the $PM_{2.5}$ concentration, we can study the causes and characteristics of air pollution more effectively. Using the relevant classification index of the daily average $PM_{2.5}$ concentration in the Technical Regulation on Ambient Air Quality Index (MEE, 2012), we divided the samples into clean (with a concentration of less than 75 μg $m^{-3}$), regular (with a concentration between 75 and 150 μg $m^{-3}$), and polluted (with a concentration greater than 150 μg $m^{-3}$). The $f_{nf}$ value in most samples in BJ (44 ± 8%) and LF (19 ± 2%) was lower during serious air pollution (Fig. 4), indicating that the high concentrations of aerosols in BJ and LF were more affected by fossil sources. One BJ sample had a low $f_{nf}$ value (36%) in January and another had a high $f_{nf}$ value (89%) in February. These samples were collected when the atmosphere was severely polluted and very clean, respectively. This might indicate that emissions from fossil

fuel sources are a decisive factor of air pollution in BJ. In the XA samples, when the atmosphere was clean, $f_{nf}$ decreased by 2–3%, indicating that the carbonaceous aerosol pollution may be more affected by biomass burning or secondary non-fossil sources from local emissions.

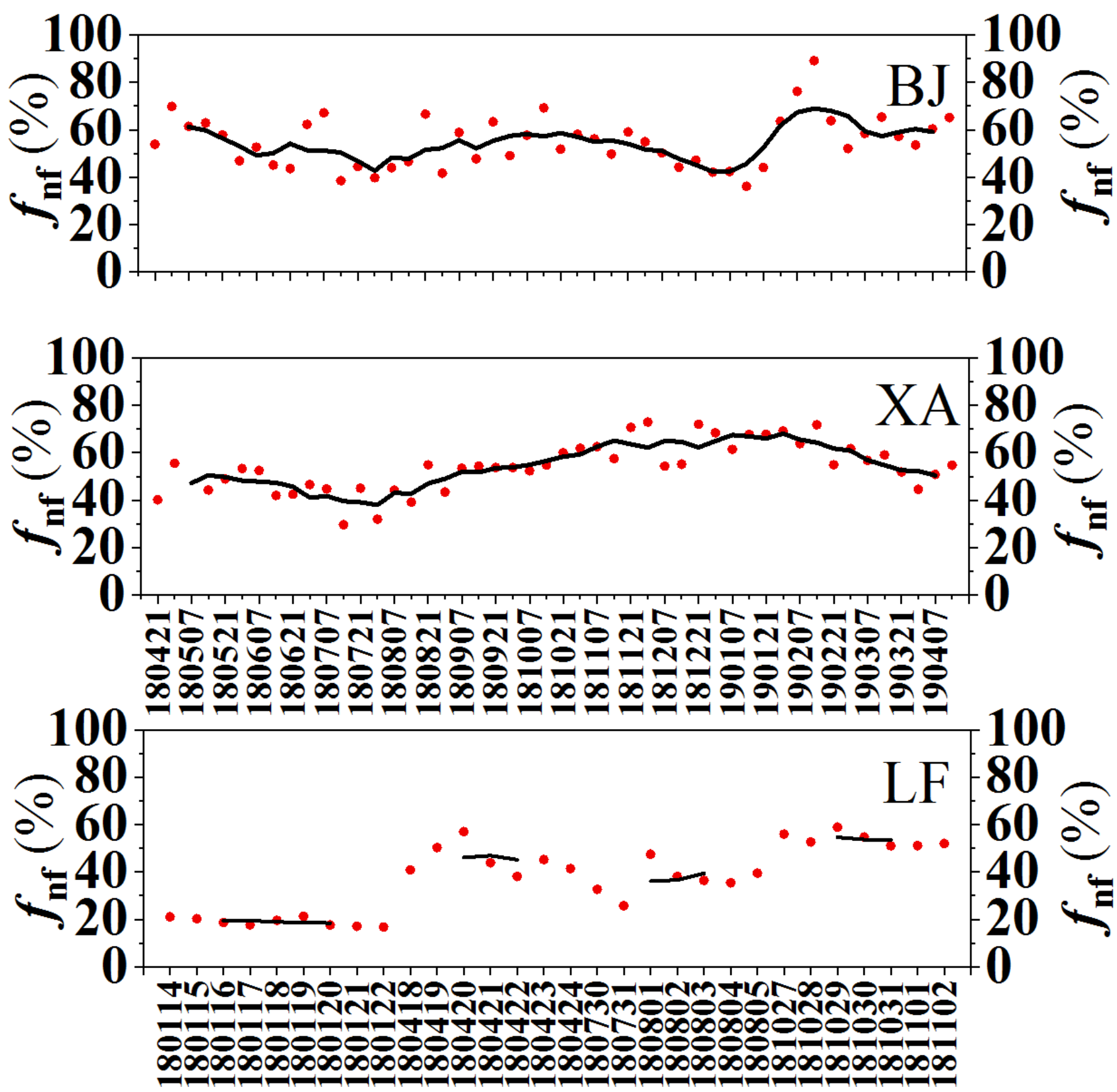

Fig. 3 Variations in proportion of non-fossil sources ($f_{nf}$) of carbonaceous aerosols at the sampling sites in Beijing (BJ), Xi'an (XA), and Linfen (LF). The red scatter dot represents the $f_{nf}$ of each sample, and the black line represents the sliding average $f_{nf}$ value of every five samples (date, "yymmdd").

Figure 5 lists the studies of $^{14}C$ in aerosols in our research area over the past few decades. With the progress of air quality management, the proportion of fossil sources

decreased by 6–17% in BJ and XA (Huang et al., 2014; Lim et al., 2020; Liu et al., 2016; Ni et al., 2018, 2020; Shao et al., 1996; Sun et al., 2012; Yang et al., 2005; Zhang et al., 2015, 2017). With the implementation of energy conservation and emission reduction policies, many non-clean fossil fuels have been transformed into clean energy. In 2019, the coal consumption in BJ was only 1.3 million tons, which was 91.5% lower than that in 2013 (BJMBS, 2020). The decline in fossil source contributions in XA (6%) was smaller than that in BJ (17%) in the past few decades. This difference can be explained as follows: the decline in coal consumption in Shaanxi Province during 2019 was not significant compared to that in 2013, whereas the consumption of liquid fossil fuels decreased by 37% (SAPBS, 2020). The particulate matter emitted from coal combustion is higher than that emitted from the combustion of liquid fossil fuels (Chen et al., 2005; England et al., 2002; Guo et al., 2014; Yan et al., 2010). Concerning non-fossil sources, China produces 939 million tons of agricultural biomass residues annually, which is the main energy source for some rural areas (Liao et al., 2004; Lu et al., 2009). In addition, the increase in urban vegetation coverage may also increase the photochemical reactions of biological volatile organic compounds (VOCs) (Gelencsér et al., 2007; NBS, 2021). Therefore, in recent years, non-fossil fuels have gradually become a major contributor to carbonaceous aerosols in BJ and XA with the reduction in the use of fossil energy.

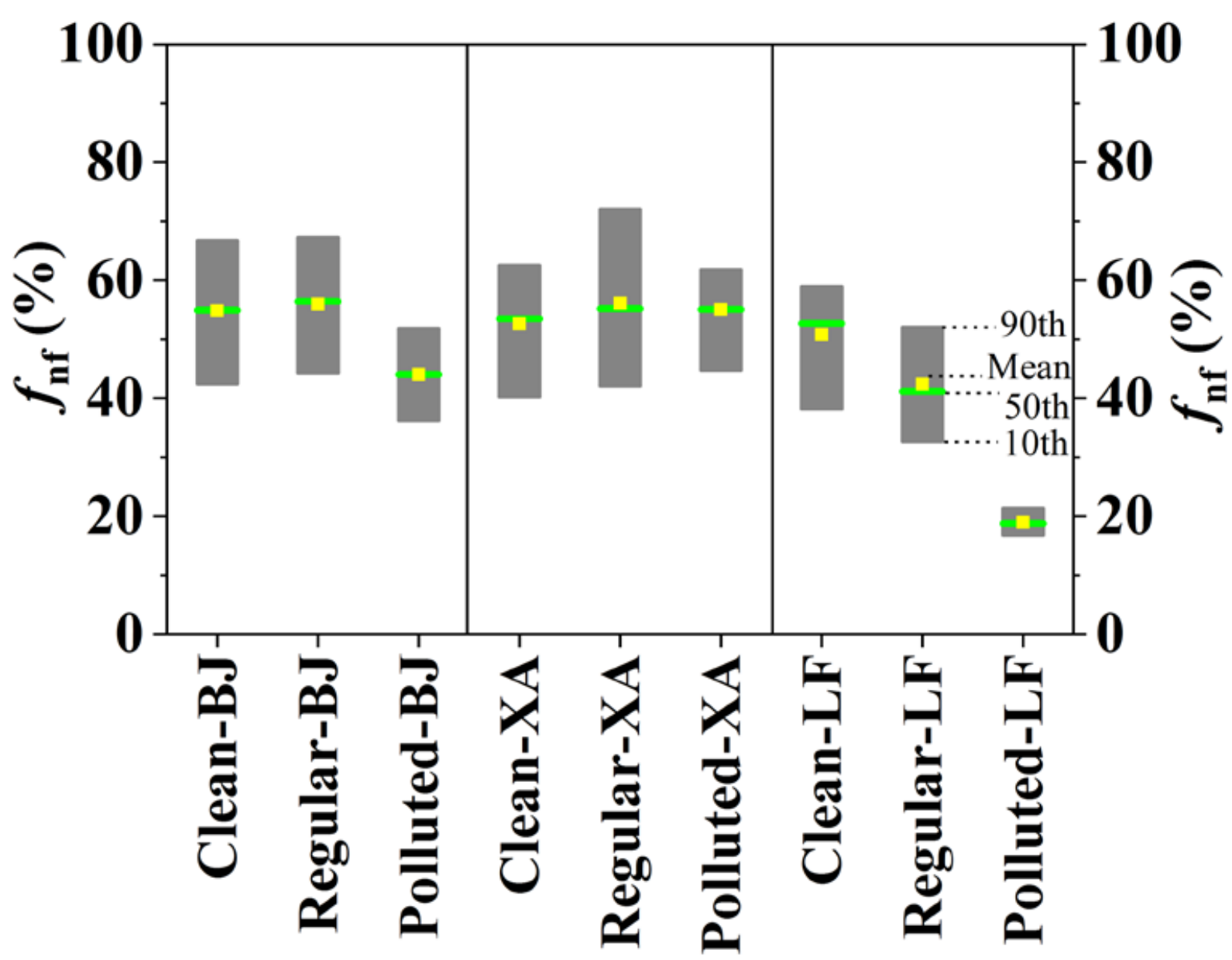

Fig. 4 Boxplot distribution of $f_{nf}$ of samples with different pollution levels. Clean samples: $PM_{2.5}$ < 75 μg $m^{-3}$; regular samples: 75 μg $m^{-3}$ ≤ $PM_{2.5}$ < 150 μg $m^{-3}$; polluted samples: $PM_{2.5}$ ≥ 150 μg $m^{-3}$.

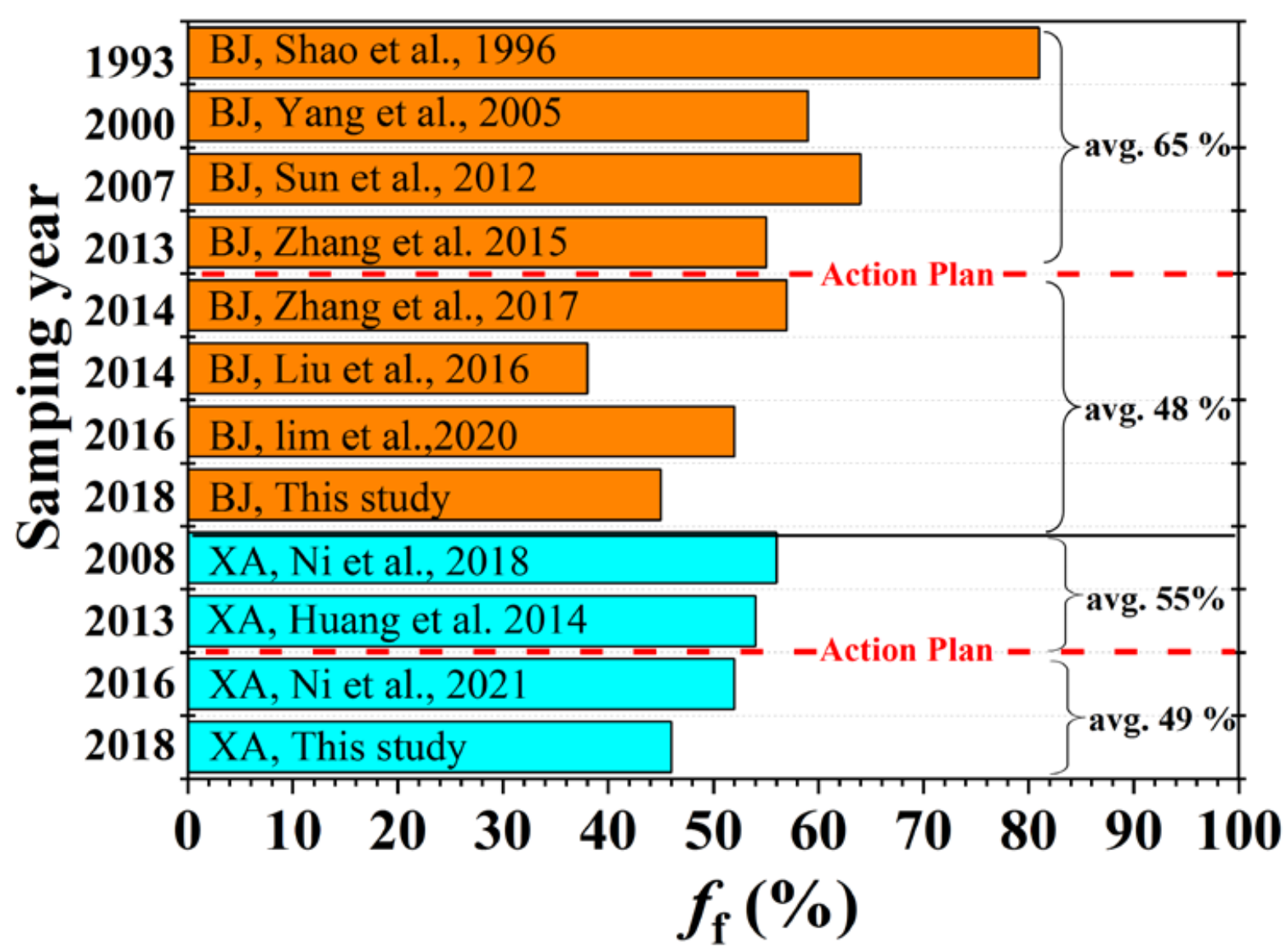

Open Access EGU

Fig. 5 Comparison of fossil proportion ($f_f$) of carbonaceous aerosol reported in different studies in Beijing (BJ) and Xi'an (XA), China. The data has been converted to the ratio of total carbon.

### 3.3 Air mass backward trajectory analysis

We analyzed and counted the backward trajectory during the sampling period; several typical types are presented in Fig. S1. Figure S1 (a) shows the type of backward trajectory with the highest frequency during the sample collection in BJ. This type of long-distance transportation from the northwest accounted for approximately 43.9% of all cases. The average $PM_{2.5}$ concentration, carbonaceous aerosol concentration, and $f_{nf}$ of the sample were 45.36 ± 22.73 μg m$^{-3}$, 9.52 ± 6.40 μgC m$^{-3}$, and 56 ± 10%, respectively. As shown in Fig. S1 (b), when air mass was transported from the south or stayed for a long time in the Hebei province, air pollution was usually more serious. These cases accounted for approximately 26.3% of all cases. The average concentrations of $PM_{2.5}$ and carbonaceous aerosols were 97.30 ± 43.61 μg m$^{-3}$ and 15.60 ± 7.94 μgC m$^{-3}$, which were 2.1 and 1.6 times of those in the northwest, respectively. The aerosol concentration of air masses transported from the southern region was higher than that from the northern regions. The $f_{nf}$ value in these cases was 46 ± 5%, which was 10% higher than in the northwest cases. Thus, air pollution in BJ might be affected by fossil sources in the Hebei province and other southern regions.

The $PM_{2.5}$ and carbonaceous concentrations were low when the air mass was transported from the northwest for a long distance at the XA site (Fig. S1 (c)). In this case, the average $PM_{2.5}$ concentration, carbonaceous aerosol concentration, and $f_{nf}$ of the samples were 93.05 ± 65.1 μg m$^{-3}$, 17.37 ± 9.61 μgC m$^{-3}$, and 62 ± 7%,

respectively. However, when air masses circulated in the Guanzhong Basin owing to topographical problems or converged into the basin from multiple directions (Fig. S1 (d)), the concentration of carbonaceous aerosol was usually high. The proportion of this type of air mass transportation accounted for 53.6% of the total cases. The average $PM_{2.5}$ concentration, carbonaceous aerosol concentration, and $f_{nf}$ of the corresponding samples were 131.95 ± 72.75 μg m$^{-3}$, 19.69 ± 10.43 μgC m$^{-3}$, and 58 ± 9%, respectively. Thus, air pollution in XA was mainly affected by the diffusion environment. The air mass remained in the upper part of the Guanzhong region for a long time when the diffusion environment was poor, causing secondary reactions and air pollution. Moreover, when the air mass came from eastern cities (e.g., Henan or Hubei provinces), $f_{nf}$ was 47%, which was significantly lower than that in other cases. This indicated that fossil source emissions in Henan and other eastern regions might contribute to air pollution in XA.

As shown in Fig. S1 (e), when the air mass was long-distance transported to the LF, the concentration of carbonaceous aerosols was relatively stable. However, pollutants accumulated when the air mass returned over and around the city (Fig. S1 (f)). In these cases, the concentrations of $PM_{2.5}$ and carbonaceous aerosols of the sample increased by 46.35–57.10%, and $f_{nf}$ decreased by 5%. Thus, the LF samples were more susceptible to the diffusion environment and the proportion of fossil sources discharged locally.

Air pollution in BJ was more susceptible to the impact of transportation from the southern region, whereas XA and LF were more affected by local emissions and diffusion environments.

### 3.4 Best estimate of source apportionment of TC using $^{14}C$ and $^{13}C$

The $\delta^{13}C$ values at the sampling sites in BJ, XA, and LF were −25.83 ± 0.81‰, −27.12 ± 0.91‰, and −23.94 ± 0.32‰, respectively. Figure 6 shows the $\delta^{13}C$ values of the samples from each city and various sources. Specifically, $\delta^{13}C$ had lower values in the BJ and LF samples during summer (−26.31 ± 0.49‰ and −25.25 ± 0.25‰, respectively) and higher values during winter (−25.17 ± 0.79‰ and −23.94 ± 0.16‰, respectively). Conversely, the lower and higher $\delta^{13}C$ values in the XA samples appeared in winter (−27.59 ± 0.443‰) and spring (−26.54 ± 1.23‰).

Compared with the existing isotope indicators of various sources (Fig. 6), the increase in $\delta^{13}C$ in the BJ and LF samples during winter may be more related to the increase in coal combustion from local and the surrounding cities. The increase in $\delta^{13}C$ in XA samples during autumn and winter may be related to the use of C4 plant fuel, whereas the decrease during winter may be related to vehicle emissions and the use of C3 plant fuels, such as wheat straw or wood.

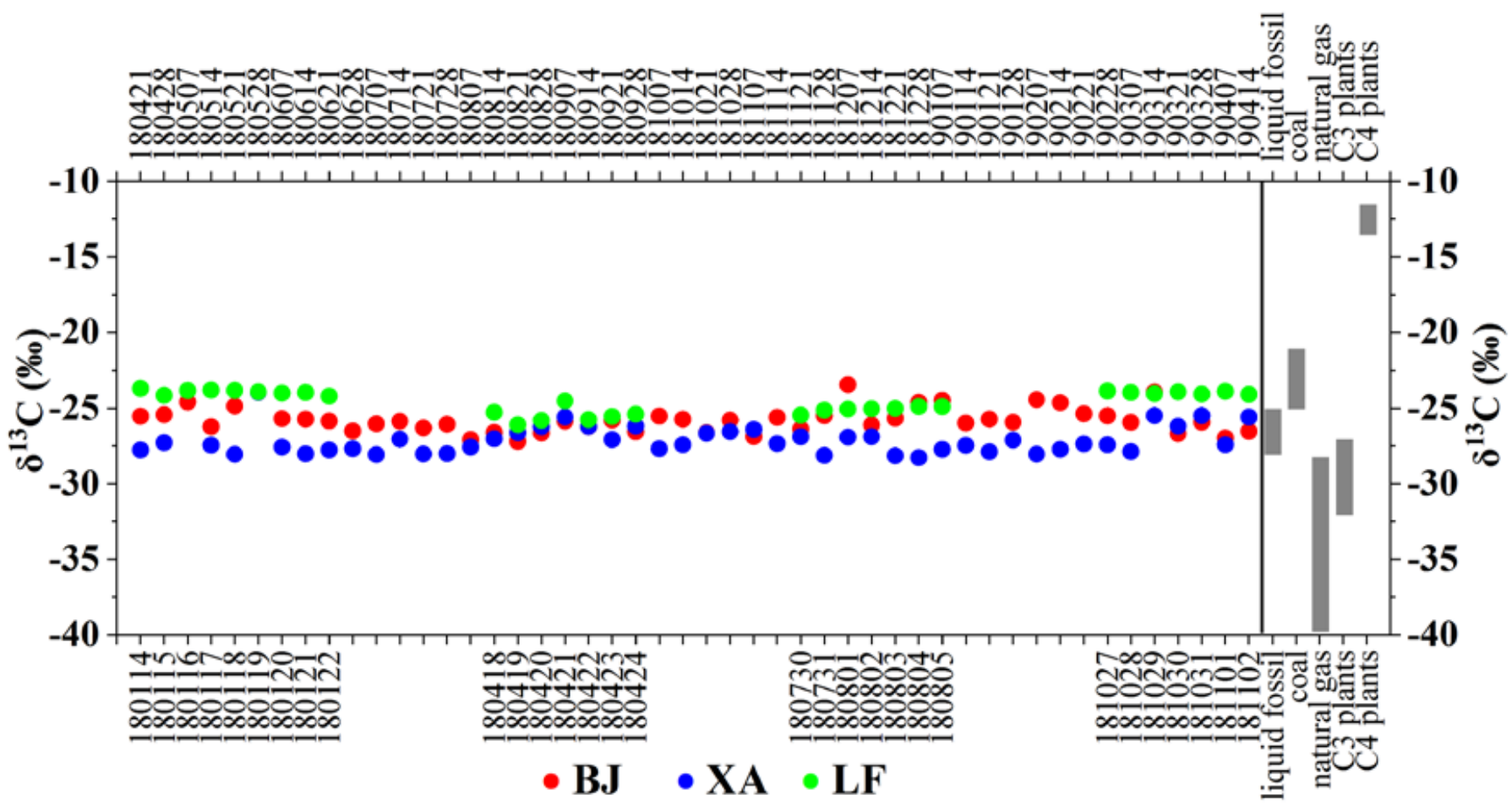

Fig. 6 $\delta^{13}C$ values of samples from Beijing (BJ), Xi'an (XA), and Linfen (LF), and comparison with the $\delta^{13}C$ distribution of various sources. The abscissa represents the sampling date (yymmdd). The gray box indicates the $\delta^{13}C$ of the main source

(Agnihotri et al., 2011; Huang et al., 2006; Lopez-Veneroni, 2009; Martinelli et al., 2002; Moura et al., 2008; Smith & Epstein, 1971; Widory, 2006).

$^{14}C$ and $^{13}C$ were used to quantify the sources of TC in the carbonaceous aerosols (Fig. 7). For the carbonaceous aerosols in BJ and XA, the best estimate of source apportionment showed that the contributions of liquid fossil fuels were 33.6 ± 12.9% and 26.6 ± 16.4%, respectively, which were greater than the contribution of coal (11.2 ± 9.1% and 19.2 ± 12.3%, respectively). In 2019, coal accounted for only 2.6% of all fossil fuels used in BJ (BJMBS, 2020). This indicates that the local combustion of coal was very low, and the coal contribution might be somewhat related to transportation from the surrounding regions. Moreover, the higher contribution of liquid fossil fuels in BJ was due to the high number of motor vehicles (6.4 million), which was 1.7 times higher than that in XA in 2019(BJMBS, 2020; XAMBS, 2020). Figure S2 shows some studies on the source apportionment of coal and liquid fossil fuels in aerosols in BJ over the past few decades. The coal contribution in BJ decreased, whereas liquid fossil fuels gradually became the main source of fossil fuels. After the implementation of the Action Plan, the proportion of coal in fossil sources decreased by approximately 32% in BJ (Gao et al., 2018; Li et al., 2013; Liu et al., 2014; Shang et al., 2019; Song et al., 2006; Tian et al., 2016; Wang et al., 2008; Zhang et al., 2014).

In contrast, coal combustion contributed 39.2 ± 20.5% to LF samples, which was greater than the contribution of liquid fossil emissions (24.6 ± 13.4%) and significantly higher than those in BJ and XA. Especially in winter, coal contributed as much as 66.2 ± 3.6% (57.0 ± 9.7 μgC $m^{-3}$). According to the data released by the Shanxi Provincial Bureau of Statistics, coal consumption in Shanxi Province was as high as 349.06 million tons in 2019, which was 46.7 times of the consumption of

liquid fossil fuels, accounting for 70.3% of the total fossil fuel consumption (SPBS, 2020). The high contribution of coal combustion in winter might be related to the use of household coal for heating by rural residents in Shanxi. This is because household coal can emit a large amount of carbonaceous particles and is an important source of carbonaceous aerosols in rural areas in northern China (Chen et al., 2005; Shen et al., 2010; Streets et al., 2003; Zhi et al., 2008).

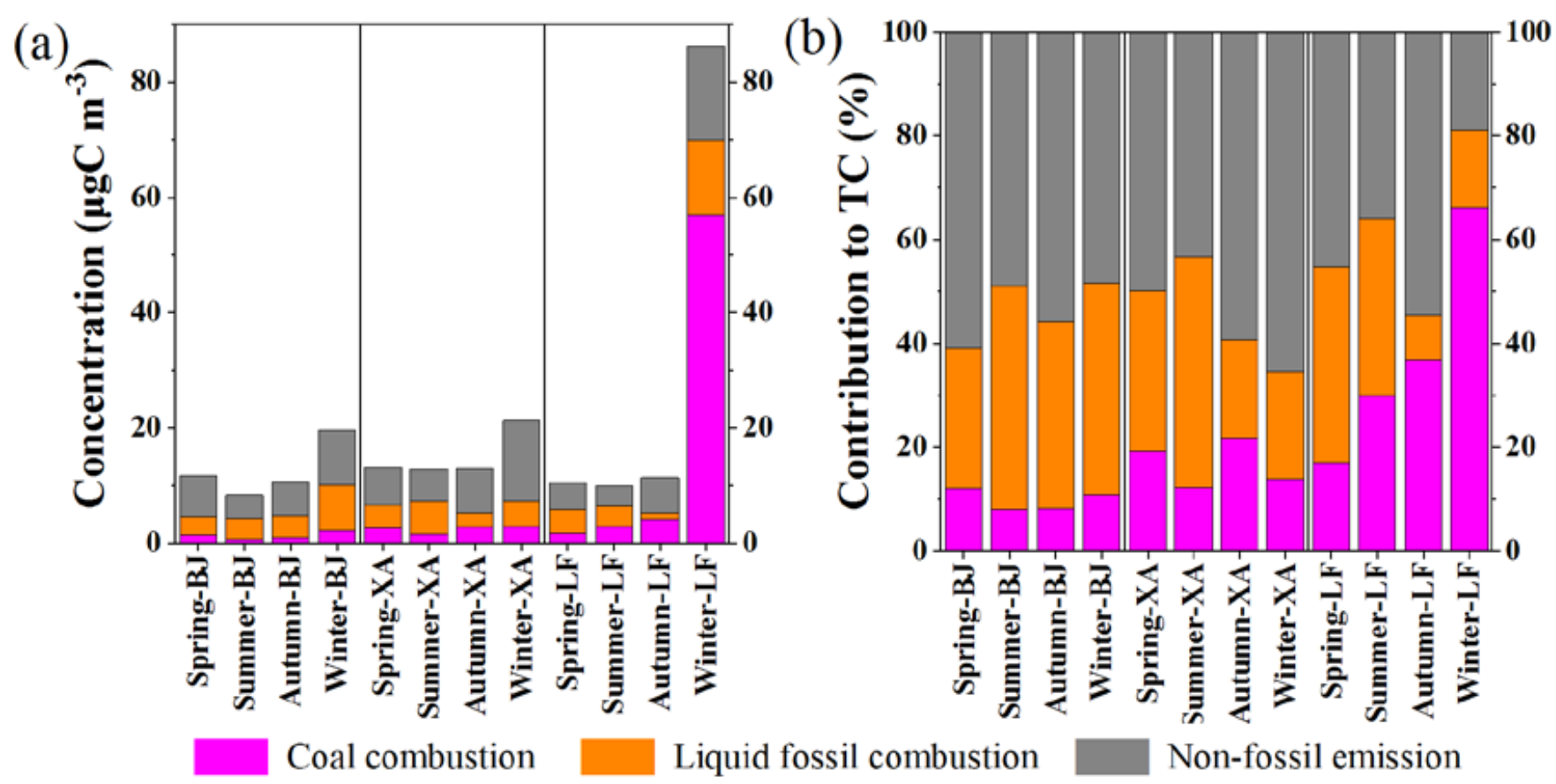

Fig. 7 Source apportionment of carbonaceous aerosols using radiocarbon ($^{14}C$) and stable carbon ($^{13}C$) isotopes at the sampling sites in Beijing (BJ), Xi'an (XA), and Linfen (LF) during different seasons. Red, blue, and orange represent the concentrations and contributions of coal combustion, liquid fossil fuel, and non-fossil sources emissions, respectively.

### 3.5 Best estimate of source apportionment of OC and EC by $^{14}C$ and Lev

The concentration of each carbon component in BJ, XA, and LF was calculated based on the combination of Lev and $^{14}C$. The best estimate of source apportionment showed in Fig. 8. The contributions of $OC_{other}$ (43.6 ± 12.9%), $OC_{ff}$ (25.5 ± 11.7%), and $EC_{ff}$ (20.5 ± 6.5%) were relatively high in BJ. The $OC_{bb}$ (23.0 ± 17.3%) and $OC_{ff}$

(39.7 ± 9.7%) were the highest contributors in XA. The LF samples showed different characteristics, and the contribution of fossil sources was significantly high, especially for the $OC_{ff}$ (56.1 ± 11.9%).

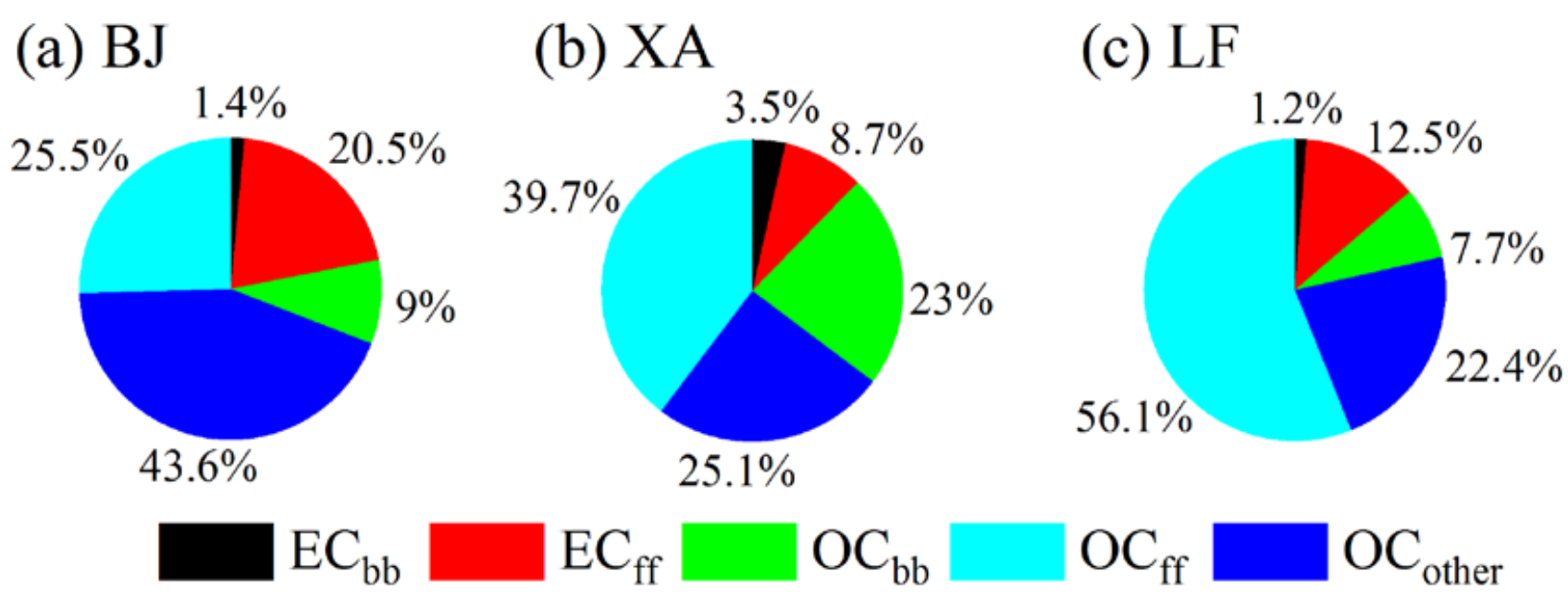

Fig. 8 Percentage of elemental carbon from biomass burning ($EC_{bb}$) and fossil-fuel combustion ($EC_{ff}$) and percentage of organic carbon from biomass burning ($OC_{bb}$), fossil-fuel combustion ($OC_{ff}$), and other sources ($OC_{other}$) for the $PM_{2.5}$ samples in Beijing (BJ), Xi'an (XA), and Linfen (LF).

**3.5.1 Biomass burning contribution to TC**

The concentrations (0.22 ± 0.27 μgC $m^{-3}$) and contributions (1.4 ± 1.1%) of $EC_{bb}$ in BJ were relatively low during the whole year (Fig. 9). The $EC_{bb}$ at the XA and LF sites had high concentrations in autumn (0.62 ± 0.42 μgC $m^{-3}$ and 0.58 ± 0.09 μgC $m^{-3}$) and winter (1.35 ± 0.60 μgC $m^{-3}$ and 1.53 ± 0.21 μgC $m^{-3}$) and low concentrations in summer (0.12 ± 0.05 μgC $m^{-3}$ and 0.05 ± 0.02 μgC $m^{-3}$), respectively. The $OC_{bb}$ concentrations in the BJ, XA, and LF samples showed an increase in autumn (1.73 ± 1.57 μgC $m^{-3}$, 3.97 ± 2.62 μgC $m^{-3}$, and 3.63 ± 0.55 μgC $m^{-3}$) and winter (2.75 ± 2.33 μgC $m^{-3}$, 8.72 ± 3.91 μgC $m^{-3}$, and 9.57 ± 1.33 μgC $m^{-3}$), respectively. Especially in the XA samples, $OC_{bb}$ had high contributions in autumn (32.4 ± 13.9%) and winter (40.7 ± 12.4%). The contribution of biomass combustion in XA (26.5 ± 19.9%) was significantly larger than those in BJ (10.4 ± 8.3%) and LF (9.0 ± 11.0%), which was also reflected in the concentration of Lev (Fig. S3). The Lev concentration in XA

($0.35 \pm 0.40$ μg $m^{-3}$) was higher than that in BJ ($0.11 \pm 0.14$ μg $m^{-3}$) and slightly higher than that in LF ($0.31 \pm 0.33$ μg $m^{-3}$). Furthermore, the Lev concentration in XA during autumn and winter was up to 6.1 times higher than that during the other seasons. Especially in winter, the proportion of Lev in the TC was $4.2 \pm 2.0\%$ in XA, which was 3.7 and 4.5 times those in BJ and LF, respectively. Zhang et al. (2015) attributed this to emissions from neighboring rural regions because such areas use biofuels for heating and cooking more commonly in winter.

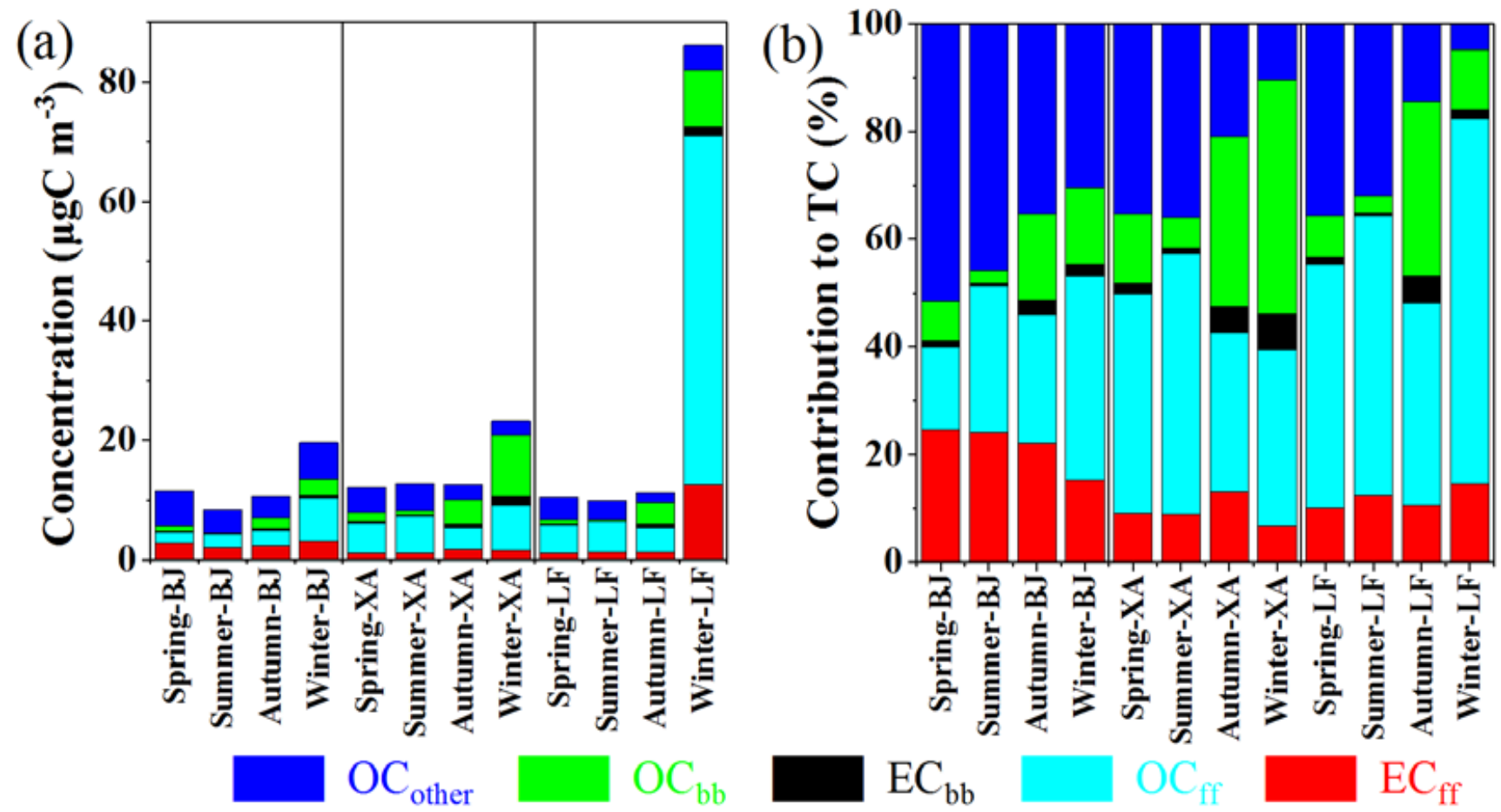

Fig. 9 Mass concentrations (μgC $m^{-3}$) (a) and percentage (b) of elemental carbon from biomass burning ($EC_{bb}$) and fossil-fuel combustion ($EC_{ff}$) and organic carbon from biomass burning ($OC_{bb}$), fossil-fuel combustion ($OC_{ff}$), and other sources ($OC_{other}$) for carbonaceous aerosols samples in Beijing (BJ), Xi'an (XA), and Linfen (LF) during different seasons.

### 3.5.2 Fossil contribution to TC

The $EC_{ff}$ concentrations at BJ (spring: $2.76 \pm 1.41$ μgC $m^{-3}$; summer: $1.99 \pm 0.84$ μgC $m^{-3}$; autumn: $2.36 \pm 2.00$ μgC $m^{-3}$; winter: $3.00 \pm 2.72$ μgC $m^{-3}$) and XA (spring: $1.14 \pm 0.73$ μgC $m^{-3}$; summer: $1.14 \pm 1.06$ μgC $m^{-3}$; autumn: $1.66 \pm 2.32$ μgC $m^{-3}$; winter: $1.48 \pm 0.73$ μgC $m^{-3}$) did not fluctuate significantly during the year. The

concentration of $EC_{ff}$ in LF during spring, summer, and autumn was relatively stable (1.05–1.24 μgC $m^{-3}$), but it was high during winter (12.66 ± 2.50 μgC $m^{-3}$), reaching 10.2 times that in summer.

The concentration of $OC_{ff}$ was slightly higher in XA during summer (6.20 ± 2.20 μgC $m^{-3}$) and winter (6.88 ± 2.38 μgC $m^{-3}$). The contribution of $OC_{ff}$ in the BJ samples increased to 31.9 ± 14.6% during winter and decreased to 17.8 ± 8.4% during spring. The $OC_{ff}/EC_{ff}$ ratios in BJ and LF during winter were approximately 2.2 ± 1.2 and 4.7 ± 0.7, respectively, suggesting that the fossil source secondary carbonaceous aerosols were higher in winter. This could be explained by the lower temperature in the winter altering the gas–particle equilibrium, suggesting that a larger portion of the $OC_{ff}$ during winter was secondary aerosol (Genberg et al., 2011). $OC_{ff}$ in LF had high concentrations in winter (58.29 ± 9.34 μgC $m^{-3}$) and low concentrations in summer (5.15 ± 1.23 μgC $m^{-3}$). This indicated that the burning of fossil sources was an important source of OC in BJ ($OC_{ff}$: 31.9 ± 14.6%) and LF ($OC_{ff}$: 67.6 ± 1.8%) during winter. Fang et al. (2017) found that fossil fuels contributed significantly (> 50%) to carbon components in the haze in East Asia during January 2014, suggesting that the aerosol contribution was generally dominated by fossil combustion sources. Therefore, using cleaner energy and cleaner residential stoves to reduce and replace the high-emission end-use coal combustion processes and control the emissions from liquid-fossil-fueled vehicles in megacities should be beneficial to the air quality.

### 3.5.3 Other non-fossil contributions to OC

In addition to the OC directly emitted from fossil and biomass fuels, there are many components of OC, such as SOC, whose source is difficult to identify. Residential oil fume emissions from urban residents, emissions from biological sources, and secondary bio-organic aerosols generated by the secondary reaction of

biomass fuels are also important components of OC (Gelencsér et al., 2007; Zhang et al., 2015).

The concentration of $OC_{other}$ in the LF samples did not vary greatly during spring (3.74 ± 1.19 μgC $m^{-3}$), summer (3.17 ± 0.53 μgC $m^{-3}$), and winter (4.06 ± 2.55 μgC $m^{-3}$), but it was lower in autumn (1.61 ± 0.20 μgC $m^{-3}$). In BJ, the contribution of $OC_{other}$ was high during spring (51.0 ± 9.3%), and its concentration was relatively high during winter (5.96 ± 5.48 μgC $m^{-3}$). Zhang et al. (2015) mainly attributed the presence of $OC_{other}$ in northern China to SOC formation from non-fossil, non-biogenic precursors. In general, secondary bio-organic aerosols in spring and autumn are mainly caused by biological emissions or long-distance transportation of biological VOCs and secondary organic aerosols (SOAs) in particulates (Gelencsér et al., 2007; Jimenez et al., 2009). The high concentration in winter may be because low temperatures drive condensable semi-volatile organic compounds (SVOCs) into the particulate phase (Simpson et al., 2007; Tanarit et al., 2008).

The $OC_{other}$ contributions in XA were high in spring (33.9 ± 7.5%) and summer (34.9 ± 10.1%). Some SOAs are formed by VOCs that are produced by burning wood or biofuels (e.g., ethanol), and they increase the load of these sources on organic aerosols (Genberg et al., 2011). Furthermore, SOC formation from these non-fossil VOCs may be enhanced when they are mixed with other pollutants, such as VOCs and $NO_x$ (Hoyle et al., 2011; Weber et al., 2007). Motor vehicles are one of the main anthropogenic sources of VOCs and $NO_x$ (Barletta et al., 2005; Liu et al., 2008). We found that motor vehicle emissions were higher in BJ and XA during winter and summer, respectively (Fig. 7), which might explain the high concentration of $OC_{other}$ in BJ during winter and in XA during summer. Huang et al. (2014) found that severe haze pollution was largely driven by secondary aerosol formation, and non-fossil

SOAs dominated, accounting for 66 ± 8% of the SOAs in XA despite extensive urban emissions. Ni et al. (2020) also considered that non-fossil sources largely contributed (56%) to SOC in XA. Thus, the control of biomass burning activities could be an efficient strategy for reducing aerosols, especially in XA.

### 3.6 Uncertainty analysis

The results of the uncertainty analysis of the given set (Table 1) of the parameters in the three cities are shown in Fig. 10. Each curve represents the probability distribution of the sources of carbon components that contribute to the TC, from which the uncertainty of the source allocation can be derived. Some results were uncertain because the input parameters of the LHS calculation varied greatly. The contributions of $OC_{ff}$ and $OC_{other}$ to the TC were mostly uncertain. This is mainly related to the uncertainty of the two parameters, Lev/$OC_{bb}$ and $(EC/OC)_{bb}$. Both these parameters depend on the burning conditions and type of biomass, as mentioned in Section 2.9. More reliable data would be obtained if $^{13}C/^{14}C$ could be performed on the pure OC fractions of the samples, which has been proven to be feasible (Huang et al., 2014; Szidat et al., 2004, 2006; Zhang et al., 2015). Other contributions have single peaks, which proves that the results of the source analysis are reliable. These results demonstrate that we can identify the main contributors.

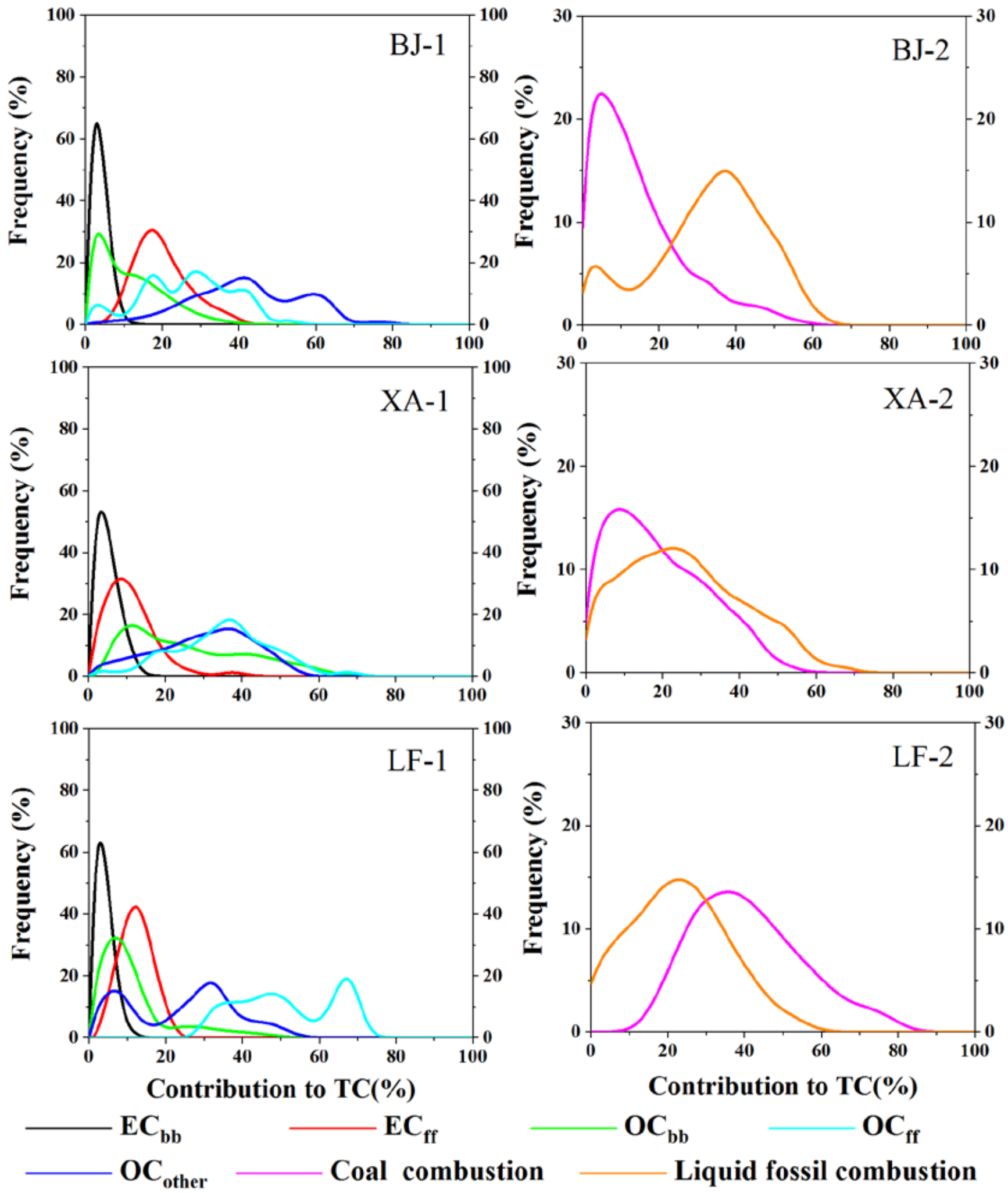

Fig. 10 Latin hypercube sampling of frequency distributions of the source contributions to total carbon (TC) from fossil, organic carbon (OC), and elemental carbon (EC) source categories (Table 1) for the samples collected in Beijing (BJ), Xi'an (XA), and Linfen (LF).

## 4 Conclusions

$PM_{2.5}$ samples were collected from BJ, XA, and LF in northern China from

January 2018 to April 2019. The main objective of this study was to quantify the sources of carbonaceous aerosols by measuring the EC, OC, Lev, $^{13}C$, and $^{14}C$ combined with LHS.

The TC accounted for approximately 17.5 ± 6%, 21.5 ± 21%, and 17.8 ± 7.2% of $PM_{2.5}$ in the samples from BJ, XA, and LF, and the corresponding concentrations were 12.50 ± 11.79 μgC $m^{-3}$, 14.64 ± 7.52 μgC $m^{-3}$, and 35.66 ± 36.53 μgC $m^{-3}$, respectively. The concentrations at the three sites showed high values in winter and low values in summer. Based on backward trajectory analysis, we found that carbonaceous aerosols in BJ were more susceptible to transportation from the southern regions. Local emissions and the diffusion environment significantly impacted carbonaceous aerosols in XA and LF.

The best estimate of source apportionment of the fossil components in the TC showed that the contribution of liquid fossil fuel combustion was 33.6 ± 12.9% and 26.6 ± 16.4% in BJ and XA, respectively, which was greater than the contribution of coal combustion (11.2 ± 9.1%; 19.2 ± 12.3%). In contrast, coal combustion contributed 39.2 ± 20.5% in LF, which was greater than the contribution of liquid fossil fuel combustion (24.6 ± 13.4%).

The best estimate of source apportionment of OC and EC indicated that the contributions of $EC_{ff}$ (20.5 ± 6.5%), $OC_{ff}$ (27.8 ± 11.7%), and $OC_{other}$ (43.6 ± 12.9%) were relatively high in BJ. The $OC_{ff}$ contribution was higher in winter (31.9 ± 14.6%), and it was 3.4 times higher than that in other seasons. The contribution of $OC_{bb}$ (23.0 ± 17.3%) and $OC_{ff}$ (39.7% ± 9.7%) was higher in XA. The contribution of biomass burning to the TC was as high as 46.94 ± 14.14% in winter. The contribution of $OC_{ff}$ in LF was significantly high (56.1 ± 11.9%), especially in winter (67.6 ± 1.8%).

The decline (6–17%) in the contribution of fossil sources since the

implementation of the Action Plan indicates the effectiveness of air quality management. In the future, the government needs to further regulate and control emissions from motor vehicles in megacities such as BJ and XA. The cleaner use of coal must be further strengthened in coal-based cities such as LF in the eastern part of the Fenwei Plain. This study indicates that attention should be paid to the control of biomass burning in northern China, especially in the Guanzhong region.

***Code and data availability:*** The data products in this paper are available at the East Asian Paleoenvironmental Science Database, National Earth System Science Data Center, National Science & Technology Infrastructure of China (http://paleodata.ieecas.cn/index_EN.aspx).

***Author contributions:*** HZ performed the data analysis and wrote the initial draft of the manuscript. ZN and WZ conceived the project and reviewed the paper. ZN and SW provided the samples. HZ, XF, SW, XL and HD conducted the measurements. All authors made substantial contributions to this work.

***Competing interests:*** The authors declare that they have no conflict of interest.

***Acknowledgments:*** The authors acknowledge the help of anonymous reviewers for improving this article.

***Financial support:*** The study was financially supported by the Strategic Priority Research Program of the Chinese Academy of Sciences (XDA23010302), National Research Program for Key Issues in Air Pollution Control (DQGG0105-02), the

National Natural Science Foundation of China (41773141, 42173082, and 41730108), Natural Science Foundation of Shaanxi province (2014JQ2-4018), Key projects of CAS (ZDRW-ZS-2017-6), and Natural Science Basic Research Program of Shaanxi Province (2019JCW-20).

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
