# Peer review of "Measurement report: Source apportionment of carbonaceous aerosol using"

_Atmospheric Chemistry and Physics, 2021_

## Author Comment (AC1)

Thank the reviewer for the time and effort on our manuscript. We appreciate the reviewer's instructive comments and suggestions. We have responded to these in the attached file, and included point-by-point responses to the reviewer's comments for consideration.

**General Comment:**

This paper investigated the source apportionments of carbonaceous aerosols collected in three northern cities in China using the measurements of carbon isotopes ($^{13}C$ and $^{14}C$) and levoglucosan. I understand the sampling campaigns and the measurements are not easy and the presented dataset should be informative. However, to be honest, I cannot find exciting things in this study and there are many technical problems needed to be addressed.

**Response:**

Thank you for your comments! The comments are very meaningful and useful. We have checked the relevant information in detail, carried out correction and reanalysis of the data. We corrected the Lev concentrations and $\delta^{13}C$ values of sample, and adjusted the end member of the parameter used in source apportionment. The correction in sample $\delta^{13}C$ caused a change about 4% for the results of source apportionment. After the correction of Lev concentrations and end member of parameter, different sources produced a change of 0.4-4.7%. These relatively small changes did not alter the main conclusions of our study. The detailed responses are shown as follows:

**Specific Comment:**

1. Line 32, Line 48 and Line 179: "stable carbon" → "stable carbon isotope"

**Response:**

Thanks for your suggestion! We have changed the corresponding words in the manuscript.

2. Line 68-70: Can you please simply tell readers why this tracer can accurately quantify the contribution of fossil and non-fossil sources?

**Response:**

Thank you for your suggestion! We have added simply explanation of the principle of $^{14}C$ in tracing non-fossil source in the manuscript:

"The natural radiocarbon ($^{14}C$) is completely depleted in fossil emissions due to the age of fossil fuels well above the half-life of $^{14}C$ (5730 years), whereas non-fossil sources show the similar $^{14}C$ as environment (Szidat, 2009; Heal, 2014)."

3. Line 74: add "a" between "useful" and "geochemical marker"

**Response:**

Thank you for your suggestion! We have changed the word in the manuscript.

4. Line 91: Can you provide the references to support "several decades"?

**Response:**

Thanks for your suggestion! We have added several references in the manuscript:

"Cities in northern China have been affected by severe haze for several decades (Cao et al., 2012; Han et al., 2016; Sun et al., 2006; Wang et al., 1990)."

5. Line 129-130: I don't think data from a TV news is appropriate for a scientific paper.

**Response:**

Thanks for your suggestion! We have updated the data and sources in the manuscript to make it more reliable:

"According to the pollutant data released by the National Air Quality Real-time Release Platform, Ministry of Ecology and Environment (MEE) of the People's Republic of China (http://106.37.208.233:20035/), the daily average atmospheric $SO_2$ concentration in LF exceeded 850 $\mu g\ m^{-3}$ on one day in January 2017."

6. Fig.1 This map is boring and less informative. I suggest the authors to add the PM$_{2.5}$ concentration at the map background.

**Response:**

Thank you for your suggestion! We have revised **Figure 1.**

[Figure]

Fig. 1 Locations and PM$_{2.5}$ concentration of Beijing (BJ), Xi'an (XA), and Linfen (LF). The background map shows the distribution of PM$_{2.5}$ concentrations in most of China from 2015 to 2019 (Li et al., 2021). The pink bars are the average PM$_{2.5}$ concentrations of the samples collected in this study during 2018 to 2019.

7. Line 147-150: So, how long for the sampling of each sample? 24 hours?

**Response:**

Yes, the samples were all collected continuously for 24 hours. We have added corresponding descriptions in the manuscript:

"A total of 124 24-hour (10 a.m. to 10 a.m. on the following day) PM$_{2.5}$ samples and 4 field blanks were obtained."

8. Line 179 Section 2.5: Please provide the corresponding reference regarding this $^{13}C$ analysis. Did you remove the carbonate fraction before $^{13}C$ analysis?

**Response:**

Thanks for your suggestion. We have added relevant references in the manuscript:

"The $^{13}C$ compositions were determined using a gas isotopic analyzer (Picarro G2131-i) in conjunction with an elemental analyzer (Elemental Combustion System 4010) at the Institute of Earth Environment, Chinese Academy of Sciences. Specifically, 0.2–0.4 mgC of sample has been placed in a precombusted tin capsule (6×10 mm) and the air was removed by squeezing. The samples were tested at 980 ℃ and 650 ℃ with 70–80 ml min$^{-1}$ helium as the carrier gas and 20–30 ml min$^{-1}$ oxygen as the reaction gas. The resulting gas mixture was then collected in Gas Isotopic Analyzer (Bachar et al., 2020)."

Yes, carbonate has been removed before testing by spraying with 1 mol L$^{-1}$ hydrochloric acid. We have added this in the manuscript.

9. Line 193-194: How to remove?

**Response:**

Carbonate was removed by spraying with 1 mol L$^{-1}$ HCl solution. We have adjusted the position of this description in the manuscript to indicate that all carbonate has been removed prior before measurement.

"Carbonate has been removed from the filters by spraying with hydrochloric acid (1 mol L$^{-1}$) before measurement."

10. Line 214-215: need to eliminate the influence of nuclear bomb excess. This correction is associated with the sampling time.

**Response:**

Thanks for your comment! We have noticed that and explained why the effect of nuclear bomb excess has been excluded in this study, which was shown in detail in the manuscript:

"Atmospheric nuclear bomb tests in the late 1950s and the early 1960s released a large amount of $^{14}C$, and the ratio of $^{14}C/^{12}C$ in atmospheric $CO_2$ roughly doubled in the mid-1960s (Hua & Barbetti, 2004; Levin et al., 2003, 2010; Lewis et al., 2004; Niu et al., 2021). However, $f_M$ in the atmosphere has been decreasing because of the dilution effect produced by the absorption of marine and terrestrial biospheres and the release of fossil fuels. In recent years, studies on background $^{14}CO_2$ in China and other countries have shown that the $f_M$ value in the atmosphere has decreased and approached 1 (Hammer et al., 2017; Niu et al., 2016). This means that the impact of the nuclear explosions has almost disappeared for current atmosphere, and the change in current atmospheric $^{14}C$ was mainly influenced by the regional natural carbon cycle and fossil fuel $CO_2$ emissions. Thus, the $f_M$ values were not corrected in this study, because the material used for biomass burning in China was mainly from crop straw (Fu et al., 2012; Street and Yarber, 2003; Yan et al., 2006; Zhang et al., 2017b), and the influence of atmospheric nuclear bomb test has basically vanished for the annual plants."

11. Line 224: This equation is questionable. A large part of OC in the air is formed by the oxidation of gaseous compounds, while some OC would be decomposed. It is impossible to make a source apportionment for OC or TC using the measurements of stable carbon isotope due to the unavoidable occurrence of isotopic fractionation.

**Response:**

Thanks for your comment! We have consulted the references on this issue.

The formation process of OC can cause the fractionation of $^{13}C$, with a range mainly in 0.03–1.40‰ (mean 0.2‰) (Aggarwal and Kawamura, 2008; Cao et al, 2011; Ho et al., 2006; Zhao et al., 2018). Therefore, a small correction (0.2‰) was made for the $\delta^{13}C_{sample}$ before it used to estimate the contributions of coal and liquid fossil fuel combustion. The correction of $\delta^{13}C$ in TC was found to cause a relatively small change (<5%) in source apportionment. The description has been added in the manuscript:

" $f_{nf} \times \delta^{13}C_{nf} + f_{coal} \times \delta^{13}C_{coal} + f_{liq.fossil} \times \delta^{13}C_{liq.fossil} = \delta^{13}C_{sample} + \beta$ \qquad (6)

$$f_{\text{coal}} + f_{\text{liq.fossil}} = f_{\text{f}} \tag{7}$$

where $f_{\text{nf}}$, $f_{\text{coal}}$, and $f_{\text{liq.fossil}}$ represent the proportions of non-fossil source, coal and liquid fossil combustion, respectively, $\delta^{13}C_{\text{nf}}$, $\delta^{13}C_{\text{coal}}$, and $\delta^{13}C_{\text{liq.fossil}}$ represent $\delta^{13}C$ from the corresponding sources, $\delta^{13}C_{\text{sample}}$ is the $\delta^{13}C$ of the samples at each site, and $\beta$ is a small correction.

Since the formation process of OC can cause the fractionation of $^{13}C$, with a range mainly in 0.03–1.40‰ (mean 0.2‰) (Aggarwal and Kawamura, 2008; Cao et al, 2011; Ho et al., 2006; Zhao et al., 2018), a small correction (0.2‰) was made for the $\delta^{13}C$ sample used in Eq. 6."

New source apportionment results have been revised throughout the manuscript.

12. Line 239, Line 247: Two important issues need to be addressed when you try to employ Lev to calculate the contribution of biomass burning. 1) You need to eliminate the interference from the emission of non-biomass burning sources. Lev is not a unique tracer of biomass burning (Wu et al. 2021); 2) You need to consider the atmospheric degradation of Lev in the air (Li et al. 2021).

**Response:**

Thanks for your suggestions! We have reviewed the relevant references and corrected the concentration of Lev. The specific content is showed in the manuscript and **Figure S3**. The new source apportionment results have been changed in the text and figures throughout the manuscript.

"Recent studies indicated that Lev was degraded to some extent during atmospheric transportation, and about 25% of them came from other non-biomass burning sources (Hoffmann et al, 2010; Wu et al., 2021). Therefore, correction of the biomass burning source lev (Lev$_{\text{bb}}$) is required before the source apportionment:

$$\text{Lev}_{\text{bb}} = \frac{\text{Lev} \times 0.75}{p} \tag{1}$$

where $p$ (0.4–0.65) is the degradation rate of Lev, which has different characteristics in each seasons. For specific $p$ value in each season, please refer to the research of Li et al. (2021)."

[Figure]

Fig. S3 Distribution of biomass-burning source Lev concentration in carbonaceous aerosols at the sampling sites in Beijing (BJ), Xi'an (XA) and Linfen (LF).

13. Line 297-309: The problem is that the wood for burning may be 20-year-old or ever older, which means that you still need to correct the effect of nuclear bomb test.

**Response:**

Thank you for your suggestion! According to the researches about biomass type, crop straw rather than wood are mainly used as biomass fuels in China (Street and Yarber, 2003; Yan et al., 2006). Therefore, the influence of nuclear bomb test can be ignored. The description has been added in the manuscript:

"According to the researches about biomass burning type, perennial biomass fuel was less frequently used in China (Fu et al., 2012; Street and Yarber, 2003; Yan et al., 2006; Zhang et al., 2017b), the impact of nuclear explosions on $^{14}$C data can be ignored, and the $f_M(nf)$ and $f_M(bb)$ of the local station should be close to the atmospheric value."

14. Line 338-341: I would say the occurrence of biomass burning can lead to an

enhanced OC/EC ratio as well.

**Response:**

Thank you for your suggestion! We have reviewed the relevant researches and made a supplementary description in the manuscript:

"Additionally, the use of biomass fuels can also enhance the OC/EC ratio (Popovicheva et al., 2014; Rajput et al., 2011). Therefore, the high OC/EC ratio indicates that carbonaceous aerosols contained a large number of SOCs or biomass burning sources, especially in XA."

15. Fig.2. Please improve the quality of this figure. We can't see the variations. Please add a Yaxis for EC and readjust Y-axis scales.

**Response:**

Thanks for your suggestion! We have increased the axes for clearer viewing and increased the resolution of the **Figure 2**.

[Figure]

Fig. 2 Variations of elemental carbon (EC), organic carbon (OC) and their ratios in PM$_{2.5}$ at the sampling sites in Beijing (BJ), Xi'an (XA), and Linfen (LF) (date, "yymmdd").

16. Fig.5. I think this comparison is inappropriate because only few samples were performed for 14C measurements in some studies of this figure. For example, Huang et al. (2014) only analyzed 6 samples collected in the 2013 winter season, however, this paper analyzed ~ 50 samples for all seasons in Xian city.

**Response:**

Thanks for your comment! Due to the different sampling periods in studies, direct comparisons would be inappropriate. To eliminate this effect, we have changed the figure for a clear comparison of the same sampling periods. The results showed that the fossil sources in Beijing show a downward trend after the Action Plan for each season/period. The corresponding results were corrected in the manuscript.

"As can be seen in **Fig. 5**, the contribution of fossil sources in Beijing decreased by about 6-15% for the different sampling season/period after the implementation of Action Plan, based on previous studies (Fang et al., 2017; Lim et al., 2020; Liu et al., 2016a, b; Ni et al., 2018, 2020; Shao et al., 1996; Sun et al., 2012; Yang et al., 2005; Zhang et al., 2015, 2017a) and this study. Among them, fossil sources decreased significantly in autumn and winter after the Action Plan, which were 15% and 14%, respectively. The contribution of fossil sources in our study decreased by 16% in winter compared with the previous results. For the polluted and clean periods, the proportion of fossil sources reduced by 6% and 9%, respectively. With the implementation of energy conservation and emission reduction policies, many non-clean fossil fuels have been transformed into clean energy. In 2019, the coal consumption in BJ was only 1.3 million tons, which was 91.5% lower than that in 2013 (BJMBS, 2020).

Different from the results in Beijing, the proportion of fossil sources in Xi'an has not decreased significantly for each season/period (Fig. 5). This difference might be

related with a small decline (< 0.5%) in coal consumption in Xi'an during 2019 compared to 2013 (XAMBS, 2014, 2020)."

[Figure]

Fig. 5 Comparison of fossil proportion ($f_f$) of carbonaceous aerosol reported in different studies in Beijing (BJ) and Xi'an (XA), China for each season/period. The data has been converted to the ratio of total carbon. The ranges shown in the upper part of the figure are the average values of each season/period before and after the Action Plan. (a) Shao et al., 1996; (b) Yang et al., 2005; (c) Sun et al., 2012; (d) Zhang et al., 2015; (e) Liu et al., 2016a; (f) Zhang et al., 2017; (g) Liu et al., 2016b; (h) Fang et al, 2017; (i) Lim et al., 2020; (j)This study; (k) Ni et al., 2018; (l) Ni et al., 2021.

17 Line 537-538: The estimated contribution of biomass burning in EC for BJ is too low. I suggest the authors to compare this result with others using different methods (e.g., radiocarbon, bottom-up emission inventory, aethalometer). First, you need to consider the atmosphere degradation of Lev; Second, the ECbb/Lev end member used in this study probably is wrong.

**Response:**

Thank you for your comment! We have made a correction to the Lev after consulting the relevant references, and adjusted the parameter (Lev$_{bb}$/OC$_{bb}$, EC$_{bb}$/OC$_{bb}$) end member used in **Table 1** according to nearly research in northern China (Huang et al., 2014; Zhang et al., 2015). In the new source apportionment (**Fig.**

8), the contribution of $EC_{bb}$ in Beijing was 1.9%, which is not much different from the previous result (1.4%).

Other related studies were combined with our results for comparison. In the research of Zhang et al. (2015) using radiocarbon and lev, the $EC_{bb}$ contribution in Beijing was 3.5%. The study of Fang et al. (2019) using radiocarbon showed a contribution of $EC_{bb}$ about 4.1%. Zhang et al. (2017a) used the AMS-PMF model along with radiocarbon and molecular tracer to analyze the source of particular matter in Beijing, and the results showed that $EC_{bb}$ accounted for 4.9%. In the source apportionment by Zhang et al. (2017b) and Tao et al. (2016) using a combination of different molecular tracers and PMF model, the contribution of biomass burning to $PM_{2.5}$ in Beijing was around 7-16.5%, and we estimated that the contribution of $EC_{bb}$ to TC was about 1.6-4.4%. Thus, our $EC_{bb}$ result was slightly smaller than those in previous studies in Beijing, but the differences were not obvious. The low $EC_{bb}$ contribution in Beijing in this study was mainly due to the low Lev concentration (Fig. S3) ($0.15 \pm 0.17$ µg m$^{-3}$), while the Lev concentration in previous studies in Beijing were 0.20~1.26 µg m$^{-3}$ (Tao et al., 2016; Zhang et al., 2015; Zhang et al., 2017b).

The relevant data and results have been changed throughout the manuscript.

[revised manuscript text omitted]

Tao, J., Zhang, L., Zhang, R., Wu, Y., Zhang, Z., Zhang, X., Tang, Y., Cao, J. and Zhang Y.: Uncertainty assessment of source attribution of pm2.5 and its water-soluble organic carbon content using different biomass burning tracers in positive matrix factorization analysis–a case study in Beijing, china. Science of the Total Environment, 543(Pt A), 326-335,

http://dx.doi.org/10.1016/j.scitotenv.2015.11.057, 2016.

Turekian, V., Macko, S., Ballentine, D., Swap, R., and Garstang, M.: Causes of bulk carbon and nitrogen isotopic fractionations in the products of vegetation burns: laboratory studies, Chemical Geology, 152, 181-192, 1998.

Wang, X., Zhu, G., Wu, Y. and Shen, X. Chemical composition and size distribution of particles in the atmosphere in north part of Beijing city for winter and summer. Chinese Journal of Atmospheric Sciences, 14, 2, 199-206, https://doi.org/10.3878/j.issn.1006-9895.1990.02.09, 1990 (In Chinese).

Wang, Z., Bi, X., Sheng, G., and Fu, J.: Characterization of organic compounds and molecular tracers from biomass burning smoke in South China I: Broad-leaf trees and shrubs, Atmospheric Environment, 43, 3096-3102, https://doi.org/10.1016/j.atmosenv.2009.03.012, 2009.

Widory, D.: Combustibles, fuels and their combustion products: A view through carbon isotopes, Combustion Theory & Modelling, 10, 831-841, https://doi.org/10.1080/13647830600720264, 2006.

Wu, J., Kong, S., Zeng, X., Cheng, Y., Yan, Q., Zheng, H., Yan, Y., Zheng, S., Liu, D., Zhang, X., Fu, P., Wang, S.,and Qi S.: First High-Resolution Emission Inventory of Levoglucosan for Biomass Burning and Non-Biomass Burning Sources in China. Environmental Science & Technology. 55 (3), 1497-1507, https://doi.org/10.1021/acs.est.0c06675, 2021.

XAMBS (Xi'an Municipal Bureau Statistics): Xi'an Statistical Yearbook-2014 China Statistics press, http://tjj.xa.gov.cn/tjnj/2014/tjnj/indexch.htm, 2014 (last access: 31 October 2021).

XAMBS (Xi'an Municipal Bureau Statistics): Xi'an Statistical Yearbook-2020 China Statistics press, http://tjj.xa.gov.cn/tjnj/2020/zk/indexch.htm, 2020 (last access: 31 October 2021).

Yan, X., Ohara, T. and Akimoto, H.: Bottom-up estimate of biomass burning in mainland china. Atmospheric Environment, 40(27), 5262-5273. http://10.1016/j.atmosenv.2006.04.040, 2006.

Yang, F., He, K., Ye, B., Chen, X., Cha, L., Cadle, S., Chan, T., and Mulawa, P.:

One-year record of organic and elemental carbon in fine particles in downtown Beijing and Shanghai, Atmospheric Chemistry & Physics, 5, 1449-1457, https://doi.org/10.5194/acp-5-1449-2005, 2005.

Zhang, Y., Shao, M., Zhang, Y., Zeng, L., LingYan, H., Zhu, B., Wei, Y., and Zhu, X.: Source profiles of particulate organic matters emitted from cereal straw burnings, Journal of Environmental Sciences, 19, 167-175, https://doi.org/10.1016/S1001-0742(07)60027-8, 2007.

Zhang, Y., Huang, R., Haddad, I., Ho, K., Cao, J., Han, Y., Zotter, P., Bozzetti, C., Daellenbach, K. R., Canonaco, F., Slowik, J., Salazar, G., Schwikowski, M., Schnelle-Kreis, J., Abbaszade, G., Zimmermann, R., Baltensperger, U., Prévôt, A. S. H., and Szidat, S.: Fossil vs. non-fossil sources of fine carbonaceous aerosols in four Chinese cities during the extreme winter haze episode of 2013, Atmospheric Chemistry & Physics, 15, 1299-1312, https://doi.org/10.5194/acp-15-1299-2015, 2015.

Zhang, Y., Ren, H., Sun, Y., Cao, F., Chang, Y., Liu, S., Lee, X., Agrios, K., Kawamura, K., Liu, D., Ren, L., Du, W., Wang, Z., Prévôt, A. S. H., Szidat, S., and Fu, P.: High Contribution of Nonfossil Sources to Submicrometer Organic Aerosols in Beijing, China, Environmental Science & Technology, 51, 7842, https://doi.org/10.1021/acs.est.7b01517, 2017a.

Zhang, Z., Gao, J., Zhang, L., Wang, H., Tao, J., Qiu, X. Chai, F., Li, Y. and Wang, S.: Observations of biomass burning tracers in pm 2.5 at two megacities in north china during 2014 apec summit. Atmospheric Environment, S1352231017305927, http://dx.doi.org/10.1016/j.atmosenv.2017.09.011, 2017b.

Zhao, Z., Cao, J., Zhang, T., Shen, Z., Ni, H., Tian, J., Wang, Q., Liu, S., Zhou, J., Gu, J. and Shen, G.: Stable carbon isotopes and levoglucosan for pm 2.5 elemental carbon source apportionments in the largest city of northwest china. Atmospheric environment, *185*(JUL.), 253-261. https://doi.org/10.1016/j.atmosenv.2018.05.008, 2018.

---

## Author Comment (AC2)

Measurement report: Source apportionment of carbonaceous aerosol using dual-carbon isotopes (13C and 14C) and levoglucosan in three northern Chinese cities during 2018-2019

**Summary:**

The authors have conducted a yearlong study characterizing filter samples for elemental carbon, organic carbon, levoglucosan, 13C and 14C carbon isotopes in three major cities in Northeastern China. They conclude from the collected data that the Action Plan for Air Pollution Prevention Control implemented in 2013 was effective in reducing the use of fossil fuels. Overall, ambient measurements and the filter analysis they conducted was thorough, and are difficult to achieve. My only concern is the interpretation of the data and the conclusions drawn. More technical detail and deeper analysis will highlight the importance of this measurement data.

**Response:**

We are grateful to the reviewer for the time and effort on the manuscript. These comments are valuable for us to improve our paper. We made corrections to some data, and revised the corresponding results and conclusions of the source apportionment. Our responses to specific comments are given below.

**Specific comments:**

1. Levoglucosan is considered a marker for biomass burning, but it does have other sources. This should be taken into consideration in the analysis (Wu et al. First High-Resolution Emission Inventory of Levoglucosan for Biomass Burning and Non-Biomass Burning Sources in China, Environ Sci Technol, 55, 3, 1497-1507, 2021)

**Response:**

Thanks for your suggestions! We have reviewed the relevant references and corrected the concentration of Levoglucosan (Lev) from biomass burning.

"Recent studies indicated that Lev was degraded to some extent during atmospheric transportation, and about 25% of them came from other non-biomass

burning sources (Hoffmann et al, 2010; Wu et al., 2021). Therefore, correction of the biomass burning source lev (Lev$_{bb}$) is required before the source apportionment:

$$Lev_{bb} = \frac{Lev \times 0.75}{p} \tag{1}$$

where $p$ (0.4–0.65) is the degradation rate of Lev, which has different characteristics in each seasons. For specific $p$ value in each season, please refer to the research of Li et al. (2021b)."

2. Some references to consider/include in the variability of $\Delta$ 13 C of sources.
(1)Pugliese, S. C.; Vogel, F.; Murphy, J. G.; Moran, M.; Stroud, C.; Ren, S.; Zhang, J.; Zheng, Q.; Worthy, D.; Huang, L.; Broquet, G. Towards Understanding The Variability In Source Contribution Of Co2 Using High-Resolution Simulations Of Atmospheric $\Delta$13Co2 Signatures In The Greater Toronto Area, Canada.
 (2)Pugliese, S. C.; Murphy, J. G.; Vogel, F.; Worthy, D. Characterization Of The $\Delta$ 13 C Signatures Of Anthropogenic Co 2 Emissions In The Greater Toronto Area, Canada. Applied Geochemistry 2017, 83, 171 - 180.

**Response:**

Thanks for your suggestion! The first reference summarized the $\delta^{13}$C values of various sources, and the second reference measured the $\delta^{13}$C value of different fossil sources. Thus, the second reference was cited in the revised version as following.

"The $\delta^{13}$C of aerosols derived from liquid fossil fuels (gasoline and diesel oil) was approximately −31 ‰ to −25 ‰ (Agnihotri et al., 2011; Huang et al., 2006; Lopez-Veneroni, 2009; Pugliese et al., 2017; Vardag et al., 2015; Widory, 2006). The $\delta^{13}$C derived from coal combustion was relatively high, ranging from −25 ‰ to −21 ‰ (Agnihotri et al., 2011; Pugliese et al., 2017; Widory, 2006)."

3. Figure 1 could be emissions inventory map, highlighting the cities where measurements were taken. To compare what is accounted for and what the authors measure can be a valuable comparison.

**Response:**

Thank you for your suggestion! To better compare the ambient particulate pollution in the study region to other parts in China, we have revised **Figure 1** by adding the PM$_{2.5}$ concentrations for similar time periods across the country.

[Figure]

Fig. 1 Locations and PM$_{2.5}$ concentration of Beijing (BJ), Xi'an (XA), and Linfen (LF). The background map shows the distribution of PM$_{2.5}$ concentrations in most of China from 2015 to 2019 (Li et al., 2021a). The pink bars are the average PM$_{2.5}$ concentrations of the samples collected in this study during 2018 to 2019.

4. Were samples taken weekly? this wasn't clear.

**Response:**

Samples were not collected weekly. Samples from Beijing and Xi'an were collected on 4 fixed days (7$^{th}$, 14$^{th}$, 21$^{st}$, and 28$^{th}$) in a month, and samples from Linfen were collected on seven consecutive days in each season. We have revised the description in the manuscript.

"At BJ and XA, PM$_{2.5}$, samples were collected on the 7$^{th}$, 14$^{th}$, 21$^{st}$, and 28$^{th}$ of

each month from April 28, 2018, to April 21, 2019. In LF, seven consecutive days in each season were selected for sample collection, and the sampling periods were concentrated in January, April, July, and October 2018. A total of 124 24-hour (10 a.m. to 10 a.m. on the following day) $PM_{2.5}$ samples and 4 field blanks were obtained."

5. More detail in the PM2.5 sampling setup is needed to describe the type of sample obtained. How was PM2.5 sampled specifically from ambient air (presumably in the presence of PM10 and larger particles)? Were there a denuder scrubbing out gasses (O3, VOCs, NOx, etc) that could react with or condense on the particles collected on the filter? If there was any chemistry happening on the filter, it would be difficult to interpret TC/OC since oxidant concentrations have also changed over the years.

**Response:**

Thanks for your comment! The sampler was equipped with a $PM_{2.5}$ impact collector, not with a denuder system, which was common in most carbonaceous studies (e.g. Cao et al., 2003; Park et al., 2018; Wang et al., 2015). Our samples were performed the chemical analysis immediately after the collection and weighing, not stored over years. We have supplemented the detailed description of sampling as follows:

"The sampler was equipped with an impact collector to collect the particles less than 2.5 μm in aerodynamic diameter."

6. This is a suggestion, not a needed comment. The use of the acronym AMS may be confusing since it's commonly used to describe the aerosol mass spectrometer. To abbreviate the accelerator mass spectrometer (AccMS? of ACLMS?)

**Response:**

Thanks for your suggestion! AMS is a special abbreviation for accelerator mass spectrometer, which has been widely used for several decades. Modification has not been made here, since aerosol mass spectrometer was not involved in this manuscript, and they would not be confused.

7. Minor clarification in line 203: what does MV stand for? I assume it means mV?

**Response:**

Thanks for your comment! The MV here is the voltage unit, and it is the abbreviation of Megavolt. We have changed it to the full name in the manuscript to avoid confusion.

"The graphite was pressed into an aluminum holder and measured using a 3 Megavolt AMS, with a precision of 3‰ (Zhou et al., 2006, 2007)."

8. Figure 3 it would be helpful to have 50% line to see when the dominant fraction shifts.

**Response:**

Thanks for your suggestion! We have added 50% lines to the Figure 3.

[Figure]

Fig. 3 Variations in proportion of non-fossil sources ($f_{nf}$) of carbonaceous aerosols at the sampling sites in Beijing (BJ), Xi'an (XA), and Linfen (LF). The red scatter dot represents the $f_{nf}$ of each sample, and the black solid line represents the sliding average $f_{nf}$ value of every five samples (date, "yymmdd").

9. In section 3.2, where there any reported data representing LF? is not, please highlight.

**Response:**

Thanks for your comment! Although LF has suffered from serious air pollution, it has received less attention because it is not a provincial capital city, so there is no reported data for comparison. We have added a description in the manuscript.

"Due to the less attention to LF, there is still a lack of related research of carbonaceous aerosols using radiocarbon in this city to compare."

10. Line 444: "topographic problems" can change to "due to the local topography"

**Response:**

Thanks for your suggestion! We have revised the corresponding description.

"However, when air masses circulated in the Guanzhong Basin due to the local topography or converged into the basin from multiple directions."

11. Figure 6 could have a cleaner x-axis, with datetime on a weekly scale.

**Response:**

Thanks for your suggestion! Since the samples were not collected weekly, we reduced the dates displayed in X-axis to make it more clearly.

[Figure]

Fig. 6 $\delta^{13}C$ values of samples from Beijing (BJ), Xi'an (XA), and Linfen (LF), and comparison with the $\delta^{13}C$ distribution of various sources. The abscissa represents the sampling date (yymmdd). The labels of top axis represent the date of BJ and XA, and the bottom represents the date of LF. The gray box indicates the $\delta^{13}C$ of the main source (Agnihotri et al., 2011; Huang et al., 2006; Lopez-Veneroni, 2009; Martinelli et al., 2002; Moura et al., 2008; Pugliese et al., 2017; Smith & Epstein, 1971; Vardag et al., 2015; Widory, 2006).

12. Figure 7 isn't black/white friendly. Can you change just one variable (instead of a solid color) something with hashed lines?

**Response:**

Thanks for your suggestion! We have changed the legend to make them better distinguished.

[Figure]

Fig. 7 Source apportionment of carbonaceous aerosols using radiocarbon ($^{14}$C) and stable carbon ($^{13}$C) isotopes at the sampling sites in Beijing (BJ), Xi'an (XA), and Linfen (LF) during different seasons. The blocks represent the concentrations and contributions of coal combustion, liquid fossil fuel, and non-fossil sources emissions, respectively.

13. Line 612-613: Was not clear the conclusion here since the distribution of allocated C sources for BJ in winter and summer appear the same. I think you meant Figure 9, since this discussion is about OCother.

**Response:**

Thanks for your comment! Figure 7 mentioned here was to explain that $OC_{other}$ had similar seasonal characteristics with the carbonaceous concentrations from motor vehicle emissions. In this version, we have revised some sentences which were shown in bold as follows to make it more clearly. The description of the similar seasonal characteristic between $OC_{other}$ and motor vehicle's contribution had been moved to the end of the paragraph.

"The $OC_{other}$ contribution and concentration in XA were high in summer (35.2 ± 10.0%) and winter (5.4 ± 4.2 µgC m$^{-3}$), respectively. **We assume that this excess is mainly attributed to SOC formation from non-fossil and primary biogenic particles.**"

"Furthermore, SOC formation from these non-fossil VOCs may be enhanced when they are mixed with other pollutants, such as VOCs and $NO_x$ (Hoyle et al., 2011; Weber et al., 2007). Motor vehicles are one of the main anthropogenic sources of VOCs and $NO_x$ (Barletta et al., 2005; Liu et al., 2008). **In Section 3.4, we found that the carbonaceous concentrations from motor vehicle emissions were high in XA during winter and summer (Fig. 7a), and the increasing of motor vehicle activities might partly explain the high concentration of OC$_{other}$ during the two seasons.**"

Overall, to assess whether the government led Action plan to reduce pollution was effective is important and long-term sampling is needed. The authors are doing valuable research, they just have to extend their analysis more

**Response:**

Thanks for your suggestions and comments! This is greatly helpful for us to improve this and future research. The *Action Plan* had been supported to be effective in controlling air pollution from many long-term observations of $PM_{2.5}$ in China (Cao et al., 2018; Wang et al., 2010, 2012; Zhao et al., 2011). In future, related research will be carried out in more cities, and a long-term observation is also considered.

*Reference*

Aggarwal, S. G. and Kawamura, K.: Molecular distributions and stable carbon isotopic compositions of dicarboxylic acids and related compounds in aerosols from Sapporo, Japan: Implications for photochemical aging during long-range atmospheric transport, Journal of Geophysical Research, 113, D14301, https://doi.org/10.1029/2007JD009365, 2008

Barletta, B., Meinardi, S., Rowland, F., Chan, C., Wang, X., Zou, S., Chan, L., and Blake, D. R.: Volatile organic compounds in 43 Chinese cities, Atmospheric Environment, 39, 5979-5990, https://doi.org/10.1016/j.atmosenv.2005.06.029, 2005.

Cao, J., Lee, S., Ho, K., Zhang, X., Zou, S., Fung, K., Chow, J., and Watson, J.:

Characteristics of carbonaceous aerosol in Pearl River Delta Region, China during 2001 winter period, Atmospheric Environment, 37, 1451-1460, https://doi.org/10.1016/S1352-2310(02)01002-6, 2003.

Cao, J., Cheng, Y. and Yu, C.: Urban air quality management in Xi'an. Indoor and built environment: Journal of the International Society of the Built Environment, Vol. 27(1) 3–6, https://doi.org/10.1177/1420326X17742007, 2018.

Hoffmann, D., Tilgner, A., Iinuma, Y., and Herrmann, H.: Atmospheric stability of levoglucosan: a detailed laboratory and modeling study, Environmental Science & Technology, 44, 694-699, https://doi.org/10.1021/es902476f, 2010.

Hoyle, C., Boy, M., Donahue, N., Fry, J., Glasius, M., Guenther, A., Hallar, A., Hartz, K. H., Petters, M., Petäjä, T., Rosenoern, T., and Sullivan, A.: A review of the anthropogenic influence on biogenic secondary organic aerosol, Atmospheric Chemistry & Physics, 11, https://doi.org/10.5194/acp-11-321-2011, 2011.

Huang, L., Brook, J., Zhang, W., Li, S., Graham, L., Ernst, D., Chivulescu, A., and Lu, G.: Stable isotope measurements of carbon fractions (OC/EC) in airborne particulate: A new dimension for source characterization and apportionment, Atmospheric Environment, 40, 2690-2705, https://doi.org/10.1016/j.atmosenv.2005.11.062, 2006.

Li, H., Yang, Y., Wang, H., Li, B., Wang, P., Li, J. and Liao, H.: Constructing a spatiotemporally coherent long-term PM2.5 concentration dataset over China during 1980–2019 using a machine learning approach, Science of The Total Environment, 765, 144263, ISSN 0048-9697, https://doi.org/10.1016/j.scitotenv.2020.144263, 2021a.

Li, Y., Fu, T. M., Yu, J., Feng, X., and Zeng, Z.: Impacts of chemical degradation on the global budget of atmospheric levoglucosan and its use as a biomass burning tracer. Environmental Science and Technology, 55, 8, 5525-5536, https://doi.org/10.1021/acs.est.0c07313, 2021b

Liu, Y., Shao, M., Fu, L., Lu, S., Zeng, L., and Tang, D.: Source profiles of volatile organic compounds (VOCs) measured in China: Part I, Atmospheric Environment, 42, 6247-6260, https://doi.org/10.1016/j.atmosenv.2008.01.070,

2008.

Lopez-Veneroni, D.: The stable carbon isotope composition of PM2.5 and PM10 in Mexico City Metropolitan Area air, Atmospheric Environment, 43, 4491-4502, https://doi.org/10.1016/j.atmosenv.2009.06.036, 2009.

Martinelli, L., Camargo, P., Lara, L., Victoria, R., and Artaxo, P.: Stable carbon and nitrogen isotopic composition of bulk aerosol particles in a C4 plant landscape of southeast Brazil, Atmospheric Environment, 36, 2427-2432, https://doi.org/10.1016/S1352-2310(01)00454-X, 2002.

Moura, J., Martens, C., Moreira, M., Lima, R., and Menton, M.: Spatial and seasonal variations in the stable carbon isotopic composition of methane in stream sediments of eastern Amazonia, Tellus B, 60, 21-31, https://doi.org/10.1111/j.1600-0889.2007.00322.x, 2008.

Smith, B., and Epstein, S.: Two Categories of 13C/12C Ratios for Higher Plants, Plant physiology, 47, 380-384, https://doi.org/10.1029/2006JD008158, 1971.

Park, S., Son, S. and Lee, S.: Characterization, sources, and light absorption of fine organic aerosols during summer and winter at an urban site. Atmospheric Research, 213 (NOV.), 370-380. https://doi.org/10.1016/j.atmosres.2018.06.017, 2018.

Pugliese, S., Murphy, J., Vogel, F. and Worthy, D.: Characterization of the Δ13C signatures of anthropogenic CO2 emissions in the greater Toronto area, Canada. Applied Geochemistry, 83, 171–180. http://dx.doi.org/10.1016/j.apgeochem.2016.11.003, 2017.

Vardag, S., Gerbig, C., Janssens, G. and Levin, I.: Estimation of continuous anthropogenic CO2: model-based evaluation of CO2, CO, D13C(CO2) and D14C(CO2) tracer methods. Atmospheric Chemistry & Physics, 15, 12705–12729. https://doi.org/10.5194/acp-15-12705-2015, 2015.

Wang, L., Jang, C., Zhang, Y., Kai, W., Zhang, Q., Streets, D., Fu, J., Lei, Y., Schreifels, J., He, K., Hao, J., Lam, Y., Lin, J., Meskhidze, N., Voorhees, S., Evarts, D. and Phillips, S.: Assessment of air quality benefits from national air pollution control policies in china. Part ii: evaluation of air quality predictions

and air quality benefits assessment, Atmospheric Environment, 44(28), 3442-3448. https://doi.org/10.1016/j.atmosenv.2010.05.058, 2010.

Wang, G., Cheng, S., Li, J., Lang, J., Wen, W., Yang, X. and Tian, L.: Source apportionment and seasonal variation of pm2.5 carbonaceous aerosol in the beijing-tianjin-hebei region of china. Environmental Monitoring and Assessment, 187(3), 1-13. https://doi.org/10.1007/s10661-015-4288-x, 2015.

Wang, S. and Hao, J.: Air quality management in china: issues, challenges, and options. Journal of Environmental Sciences, 24(1) 2–13, https://doi.org/10.1016/S1001-0742(11)60724-9, 2012.

Weber, R., Sullivan, A., Peltier, R., Russell, A., Yan, B., Zheng, M., Gouw, J. D., Warneke, C., Brock, C., and Holloway, J.: A study of secondary organic aerosol formation in the anthropogenic-influenced southeastern United States, Journal of Geophysical Research Atmospheres, 112, D13302, https://doi.org/10.1029/2007jd008408, 2007.

Widory, D.: Combustibles, fuels and their combustion products: A view through carbon isotopes, Combustion Theory & Modelling, 10, 831-841, https://doi.org/10.1080/13647830600720264, 2006.

Wu, J., Kong, S., Zeng, X., Cheng, Y., Yan, Q., Zheng, H., Yan, Y., Zheng, S., Liu, D., Zhang, X., Fu, P., Wang, S.,and Qi S.: First High-Resolution Emission Inventory of Levoglucosan for Biomass Burning and Non-Biomass Burning Sources in China. Environmental Science & Technology. 55 (3), 1497-1507, https://doi.org/10.1021/acs.est.0c06675, 2021.

Zhao, H. Niu, Z. and Feng, X.: Factors influencing improvements in air quality in Guanzhong cities of china, and variations therein for 2014–2020. Urban Climate, 38(4), 100877. https://doi.org/10.1016/j.uclim.2021.100877,2021.

Zhou, W., Zhao, X., Xuefeng, L., Lin, L., Zhengkun, W., Peng, C., Wengnian, Z., and Chunhai, H.: The 3MV multi-element AMS in Xi'an, China: Unique features and preliminary tests, Radiocarbon, 48, 285-293, https://doi.org/10.1016/j.atmosenv.2014.05.058, 2006.

Zhou, W., Lu, X., Wu, Z., Zhao, W., Huang, C., Li, L., Peng, C., and Xin, Z.: New

results on Xi'an-AMS and sample preparation systems at Xi'an-AMS center, Nuclear Instruments & Methods in Physics Research, 262, 135-142, https://doi.org/10.1016/j.nimb.2007.04.221, 2007.